# BEATRIX: IMPROVING OUT-OF-DISTRIBUTION GENERALIZATION OF EEG FOUNDATION MODEL VIA INVARIANT CONTRASTIVE FINE-TUNING

## ABSTRACT

The advent of large-scale foundation models has revolutionized EEG analysis; however, their ability to generalize to Out-of-Distribution (OoD) brain signals remains limited due to the inherent variability in physiological states, individual differences, and experimental setups. To address these challenges, we introduce Beatrix, a novel spectral EEG foundation model that achieves state-of-the-art OoD generalization across diverse brain activity tasks. Beatrix leverages a unique analytic wavelet-based spectral tokenization that captures the intricate non-stationary dynamics of EEG signals, and employs a semi-causal generative modeling approach during pre-training, enabling it to learn expressive latent representations capable of both interpolation and extrapolation across temporal and frequency domains. For fine-tuning, we propose an innovative Contrastive Invariant Fine-Tuning (CIFT) method that enhances domain-invariant learning without the need for explicit environment labels, thus significantly improving OoD generalizability in a parameter-efficient manner. Our multi-view Transformer architecture further integrates both spectral and temporal information, allowing Beatrix to comprehensively model EEG signals across channels. Extensive experiments demonstrate that Beatrix consistently outperforms existing EEG models in tasks such as seizure detection and forecasting, auditory neural decoding, motor imagery, and sleep staging, showcasing its robustness and broad applicability. By achieving superior performance with reduced fine-tuning costs, Beatrix represents a significant advancement in the field of EEG foundation models.

## 1 INTRODUCTION

The advent of modern neuroelectrophysiological techniques such as electroencephalography (EEG) has revolutionized our capacity to monitor neural functions with high precision, offering unprecedented insights into brain activity. These advances are particularly crucial for diagnosing neurological disorders and deepening our comprehension of brain function. Inspired by the success of foundation models in other domains, we have witnessed a surge of interest in developing analogous models for EEG analysis (Jiang et al.; Zhang et al., 2024; Yuan et al., 2024a;b; Wang et al., 2023; Yang et al., 2023). These EEG Foundation Models (EFMs), developed on time or time-frequency (spectral) domain representation of raw brain records, exhibit the potential to markedly enhance EEG-based applications such as neural decoding and brain-computer interfaces, and to refine diagnosis and treatment strategies for neurological conditions like epilepsy.

Despite their promise, the development of general-purpose and domain-specific EFMs faces significant challenges due to the inherent complexity and variability of EEG signals. These signals are affected by a wide range of factors, such as age, cognitive state, eye movements, etc.. (Croce et al., 2020). This variability is further amplified by differences in electrode setups and experimental conditions across different institutions, making it difficult for EFMs to generalize well to unseen data. Although efforts have been taken to promote robustness to distribution shifts in EEG records, it is difficult to define and partition **domains** or **environments**, a necessity explicitly or implicitly assumed for many OoD generalization techniques (Lai & Wang, 2024; Creager et al., 2021), as the EEG signal is inherently nonstationary. This challenge is especially pronounced in epilepsy-related tasks, where the diversity of seizures and the high inter- and intra-subject variabil-

ity in EEG recordings—spanning interictal, pre-ictal, and ictal phases—complicate generalization effortss (Yuan et al., 2024b; Guerrini et al., 2023; Assogba et al., 2010). Furthermore, most existing EFMs rely primarily on **temporal domain representation**, with few models leveraging **time-frequency** or spectral one. This potentially limits their ability to fully capture the complexity of EEG data, particularly in tasks where pure time-domain approaches may fall short (Ma et al., 2021).

While numerous large-scale, publicly accessible datasets support seizure detection research, a scarcity exists for the intricate cases of rare epilepsy types and seizure forecasting due to privacy concerns. Consequently, many existing EFMs (Yuan et al., 2024a; Jiang et al.; Yuan et al., 2024b) have relied on private data for pre-training and/or evaluation, complicating the advancement of further research.

To address these challenges, we present Beatrix, a pioneering spectral EFM designed for robust out-of-distribution (OoD) generalization. Beatrix is subjected to a two-stage pre-training process using the most extensive open-source EEG corpus to date, spectrally tokenizing EEG signals and disentangling time-frequency and channel interactions to capture comprehensive spatiotemporal patterns. Furthermore, we introduce a novel environment-aware fine-tuning method known as Contrastive Invariant Fine-Tuning (CIFT). This method learns domain-invariant features without the need for explicit environment information, thereby bolstering the downstream performance over challenging benchmarks out-of-disease and out-of-institute seizure-related asks. CIFT achieves this by leveraging new information from the interaction between spectrally and temporally encoded prompts, enhancing the model's robustness to distribution shifts. Remarkably, our approach substantially reduces computational costs compared to traditional full-parameter tuning. This improvement in generalization is also observed across a variety of OoD generalization baselines.

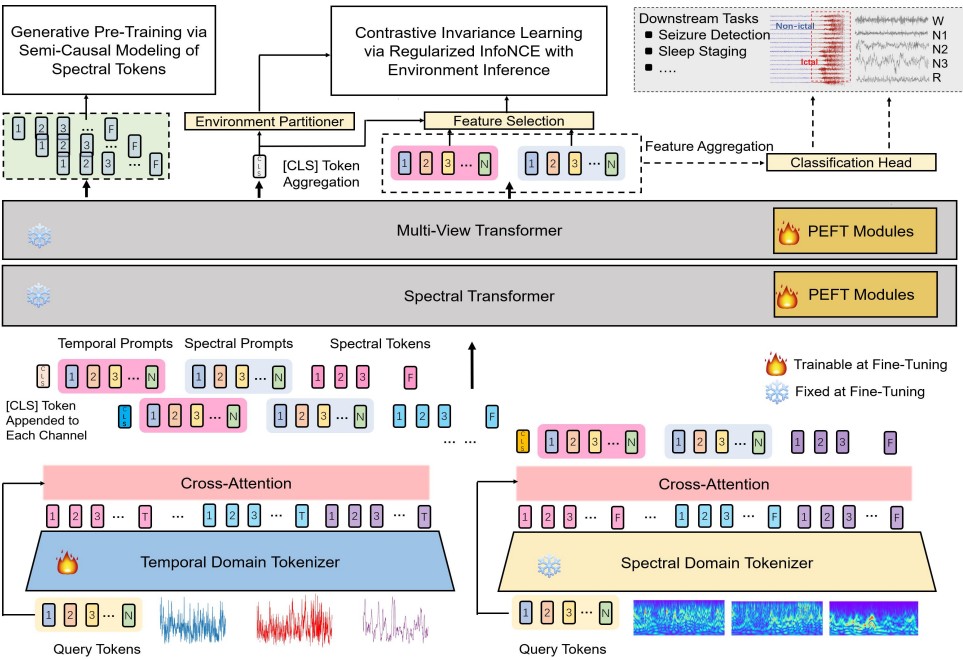

Figure 1: **Overview of Our Work**. Illustration of *Beatrix*, an EEG foundational model consists of Spetral and Multi-View Transformer, which perform self-attention on spectral token embeddings within each channel and between all channels, respectively. Firstly, it is pre-trained on spectral tokens. Secondly, it is fine-tuned on temporal and spectral tokens through invariance-aware contrastive learning with environment inference to improve OoD generalization without explicit domain partition.

Our main contributions are summarized as follows:

**A Spectral EEG Foundation Model for Epiletic and Non-epileptic Subjects** We present Beatrix, a spectral EEG foundation model pre-trained on over 32,900 hours of EEG data from both

healthy and diseased individuals. Beatrix demonstrates excellent generalizability across heterogeneous epilepsy patients and shows promise in various biomedical applications, including auditory brain decoding, motor imagery and sleep staging.

**Environment-Aware Contrastive Fine-Tuning for OoD Generalization** We propose a novel environment-aware fine-tuning method that bolsters domain-invariant representation without explicit environment information. Beatrix, fine-tuned with CIFT, achieves state-of-the-art performance in OoD seizure detection and forecasting, as well as non-epilepsy tasks such as auditory and motor imagery decoding and sleep staging.

**Spectrotemporal Integration of EEG Representation** Our ablation study indicates that the integration of spectral and temporal information during fine-tuning is crucial for significant improvements in performance.

## 2 PRELIMINARIES

### 2.1 TASK FORMULATION

We consider an EEG foundation model parameterized by $\theta$ unsupervisedly pre-trained on large-scale EEG corpus, which will be fine-tuned on various downstream datasets involving heterogeneous physiological and neurological conditions. The EEG recording of a subject is a multivariate time series $\mathbf{X}_{1:T} \in \mathbb{R}^{T \times N}$, where $T$ is the number of sampling points, and $N$ is the number of electrodes, which may alter depending on experimental settings. Given a spectral representation of EEG sample $\mathbf{X}_{1:T}$ in the time-frequency domain $\mathbf{S} \in \mathbb{R}^{T \times F \times N}$, where $F$ is the number of frequencies, and the corresponding label $y \in \{0,1\}^C$, where $C$ is the number of classes, defined and annotated by clinical electroencephalographers, the main problem of interest is OoD generalizable fine-tuning for $K$-class EEG recognition task, in which different subjects or disease subtypes can be regarded as a *domain* or *environment*. Our goal is to learn a very small proportion of fine-tunable parameters $\Delta\theta$, which is typically low-rank, so that the adapted model parameterized by $\theta + \Delta\theta$ can generalize to unseen environments.

Formally, for a given heterogeneous EEG dataset $\mathcal{D} := \{(\mathbf{X}_i, y_i) \in \mathcal{X} \times \mathcal{Y}\}_{i=1}^n$, where $\mathcal{X}$ and $\mathcal{Y}$ denote the input and target space, respectively, and the set of environment labels designated for each sample $\mathcal{E}_{\text{train}}$, which is not necessarily available during learning. We aim to learn $f$ in function space $\mathcal{F}$ parameterized by $\theta + \Delta\theta$, which is robust to distribution shifts with regard to the loss function $\ell : \mathcal{Y} \times \mathcal{Y} \times \mathcal{E}_{\text{train}} \to \mathbb{R}$ and joint distribution $\mathbb{P}_{\mathbf{X}^e, y^e}$ through a minmax optimization problem (Arjovsky et al., 2019; Lu et al., 2021)

$$\min_{f \in \mathcal{F}} \max_{e \in \mathcal{E}_{\text{train}}} \mathbb{E}_{\mathbb{P}_{X^e, Y^e}} \ell(f(x), y; e), \tag{1}$$

which is the average between the predicted and the target value $y_i$ in $e \in \mathcal{E}_{\text{train}}$.

### 2.2 RELATED WORK

**OoD Generalization in EEG-Based Applications** Out-of-Distribution (OoD) generalization in EEG-based biomedical applications is a significant challenge, particularly when dealing with heterogeneous data from diverse domains. The focus is on extracting consistent features across domains while discarding misleading ones. Wang et al. propose data augmentation techniques to address OoD scenarios (Wang et al., 2024b), while others enhance domain generalization through mutual reconstruction strategies (Wang et al., 2022b) and mutual information-based methods (Jeon et al., 2021). Yuan et al. present preprocessing techniques to improve the OoD generalizability of pre-trained models (Yuan et al., 2024b).

**EEG Foundation Models** The development of foundational models for EEG has gained momentum. Models like LaBraM (Jiang et al.), designed for general EEG analysis, and Brant (Zhang et al., 2024), tailored for intracranial signals, have shown promise in seizure detection and forecasting. PPi (Yuan et al., 2024b), pre-trained on a large SEEG corpus, demonstrates robustness to domain shifts and achieves state-of-the-art results in subject-independent seizure detection. Brant-2, building on this, incorporates both stereo- and scalep-electroencephalography modalities during pre-training (Yuan et al., 2024a).

See more related work in Appendix F.

## 3    METHODS

### 3.1    TIME-FREQUENCY REPRESENTATION AND TOKENIZATION

**Analytic Wavelet Spectral Analysis** For effective EEG signal analysis, accurately capturing the complex, non-stationary dynamics of brain activity is essential, with time-frequency representation being key. Techniques like Short-Time Fourier Transform (STFT) and Continuous Wavelet Transform (CWT) are utilized to achieve this balance. However, STFT faces limitations due to the fixed trade-off between temporal and spectral resolutions, which can result in spectrogram leakage when the window length is set, impacting the accuracy of the analysis (Wang et al., 2023). The CWT addresses these limitations by decomposing the signal into a set of dilated and translated versions of a predefined mother wavelet. This approach allows for a more favorable balance between temporal and spectral resolutions compared to STFT (Arts & van den Broek, 2022).

In this work, we employ the Analytic Wavelet Transform (AWT) Lilly & Olhede (2010), a complex-valued extension of CWT, to extract both magnitude and phase information from the time-scale or time-frequency domain. This approach is especially beneficial for non-stationary signals, where frequency content fluctuates over time. The AWT provides a more accurate estimation of instantaneous frequency and superior frequency reassignment properties than real-valued CWT and STFT. Formally, the AWT of a signal $f(t)$ with respect to a mother wavelet $\psi(t)$ is defined as

$$AWT_f(a,b) = \frac{1}{\sqrt{|a|}} \int_{-\infty}^{\infty} f(t)\psi^* \left( \frac{t-b}{a} \right) dt, \qquad (2)$$

where $a$ and $b$ are scaling and translation parameters controllinnf the scale and position of the wavelet. $\psi^*(t)$ represents the complex conjugate of $\psi(t)$.

Mother wavelets from Generalized Morse Wavelet (GMW) family, defined by its Fourier transform

$$\hat{\psi}_{\beta,\gamma}(\omega) = \int_{-\infty}^{\infty} \psi_{\beta,\gamma}(t)e^{-i\omega t}dt = c_{\beta,\gamma}\Theta(\omega)\,\omega^{\beta}e^{-\omega^{\gamma}}, \qquad (3)$$

where $c_{\beta,\gamma}$ is the normalization factor, $\Theta(\cdot)$ is the Heaviside function, and $\beta$ and $\gamma$ are two controlling parameters, are used within the scope of this work. During pre-training, we uniformly sample $\beta$ from $[1, 16]$ and $\gamma$ from $[0.5, 2.0]$ uniformly to ensure our model can handle diverse spectral representations and minimize biases resulted from wavelet shape variations. The log-transformed amplitude spectrogram of the signal undergo z-score normalization to ensure numerical stability. Unless otherwise stated, we use GMW mother wavelet with $\beta = 16, \gamma = 1$ during fine-tuning.

**Spectral Tokenizer** Previous studies (Jiang et al.; Cai et al., 2023; Yuan et al., 2024b) have investigated time-domain EEG tokenization using techniques such as Vector-Quantized Variational Autoencoders (VQVAEs) and linear projections. We propose a spectral tokenizer that transforms raw EEG signals into a rich time-frequency representation using a Vector-Quantized Generative Adversarial Network (VQGAN). This model employs a linear patch embedding layer followed by a Transformer and convolutional downsampling layers, which efficiently reduce the input spectrograms into a lower-resolution latent space. A decoder is then trained to reconstruct the input from the quantized latent vectors. The VQGAN encoder is used as the tokenizer. Further details are provided in Appendix C.1.

**Temporal Tokenizer** During the fine-tuning phase, we employ a convolutional neural network to encode temporal features from raw EEG signals, which complements the spectral domain features in our proposed contrastive fine-tuning approach. Unlike spectral tokenizer, this network is uniquely trained-from-scratch for each specific downstream dataset. Further details are provided in Appendix C.3.

### 3.2    MAIN ARCHITECTURE

Following recent work on foundation models, our network is based on Transformer. Previous EEG foundation models (Zhang et al., 2024; Yuan et al., 2024a; Jiang et al.) have primarily focused on

processing a single or fixed number of EEG channels. By contrast, Beatrix employs two distinct Transformers. The first, termed the **spectral Transformer**, captures interactions among tokens within the same channel. The second, known as the **multiview Transformer**, captures interactions among tokens across different channels. Additionally, inspired by previous work such as (Shazeer, 2020; Nguyen et al., 2024), we introduce several minor modifications: 1) **SwiGLU**. We replace the feedforward layer with SwiGLU (Shazeer, 2020), a gated linear unit with Swish nonlinearity to improve network capacity and expressivity. 2) **Feature Scaling**. Transformer's expressivity may deteriorate due to the low-pass nature of attention, which causes oversmoothing issue where token become identical as the depth grows (Nguyen et al., 2024; Wang et al., 2022a; Shi et al., 2022). Therefore, we adds a feature scaling layer (Nguyen et al., 2024) after multi-head self-attention. After being trained on the curated pre-training data corpus, the tokenizer is frozen and the main part of Beatrix is pretrained on the tokenized data. The tokenizer extract latent embeddings for each channel independently and leave modeling of the interchannel correlations to the main part of Beatrix. More details about the main architecture can be found in Appendix C.2.

### 3.3 Model Development and Downstream Adapation

**Open-source pre-training EEG corpus** We have assembled an extensive corpus, exceeding 32,900 hours, for pre-training purposes, from a diverse arrange of publicly available datasets that have been free of security and privacy issues for academic purposes. A comprehensive description of the data collection, cleansing, and preprocessing procedures can be found in Appendix E. To our knowledge, this is arguably one of the largest openly accessible EEG corpus curated specifically for pre-training.

**Two-Stage Pre-Training** We adopt a two-stage pre-training process of Beatrix, beginning with training the VQGAN tokenizer, which is then frozen while the foundational model is trained on the same EEG data corpus. As illustrated in Figure 2, unlike previous work (Jiang et al.; Zhang et al., 2024; Wang et al., 2023), we adopt a semi-causal generative modeling approach. The purpose is to enforce the model to acquire not only the ability to interpolate corrupted spectral patches but to extrapolate across both temporal and frequency dimensions as well, allowing for developing a genuinely expressive latent representation.

Formally, given an input sequence $\boldsymbol{x} = (x_1, x_2, \ldots, x_n)$, we assume there are $k$ non-causal spans $\{\boldsymbol{x}_{s_1}^{m_1}, \ldots, \boldsymbol{x}_{s_k}^{m_k}\}$, where $\boldsymbol{x}_{s_i}^{m_i} = (x_{s_i}, \ldots, x_{m_i-1})$. Within each non-causal span $x_{s_i}^{m_i}$, we randomly replace a proportion of the tokens with a special placehoder [MASK] and use bidirectional attention to obtain contextual information. Unidirectional attention is used to predict tokens autoregressively in causal spans. Negative log-likelihood of reconstructed spectrograms are used as learning goal

$$\mathcal{L}_{\mathrm{NLL}} = \mathbb{E}_{\boldsymbol{x}} \sum_{i=0}^{k} \sum_{t=m_i}^{s_{(i+1)}} \log P(x_t | x_{<t}, \{\boldsymbol{x}_{s_j}^{m_j}\}_{j<i,\mathrm{unmasked}}) P(\{\boldsymbol{x}_{s_j}^{m_j}\}_{\mathrm{masked}} | \{\boldsymbol{x}_{s_j}^{m_j}\}_{\mathrm{unmasked}}), \quad (4)$$

where $m_0 = 1, s_{(k+1)} = n$, and $\{\boldsymbol{x}_{s_j}^{m_j}\}_{j<i} = \{\boldsymbol{x}_{s_1}^{m_1}, \cdots, \boldsymbol{x}_{s_{(i-1)}}^{m_{(i-1)}}\}$. Non-causal spans and their positions are randomly sampled and do not overlap with one another. More details about pre-training are available in Appendix D.

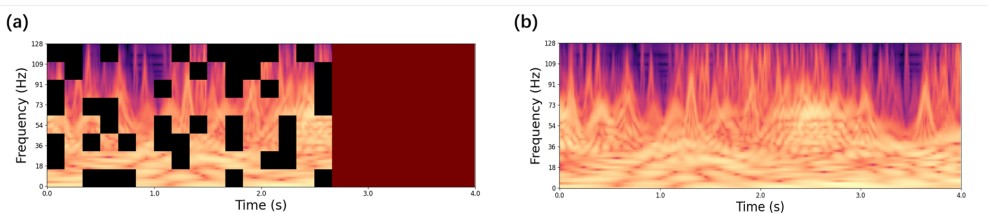

Figure 2: **Illustration of Semi-Causal Generative Modeling in Pre-training of Beatrix.** (a) Masked AWT spectrogram processed by the tokenizer and encoder, where masked tokens within the non-causal span are highlighted in black, and those within the causal span are marked in dark red. (b) Full AWT spectrogram reconstructed by the decoder, a step that is utilized during pre-training only.

**Contrastive Invariant Fine-Tuning** We design Contrastive Invariant Fine-Tuning (CIFT) to adapt our spectrally pre-trained foundation model to downstream datasets with minimal additional train-

able parameters. CIFT leverages parameter-efficient low-rank adapters to reduce computational cost while facilitating out-of-distribution (OoD) generalization.

At the core of CIFT is an automated prompt generation system powered by cross-attention mechanisms. This system devises continuous prompts based on token embeddings from both the spectral tokenizer and a complementary temporal tokenizer. The temporal tokenizer is equipped with inductive biases optimized for handling multiscale time-series data, ensuring that the prompts effectively capture the nuances of EEG signals across both spectral and temporal domains. These prompts are instrumental in guiding the main tokens, which are derived from the time-frequency representation of EEG data by the spectral tokenizer. The resulting embeddings are then utilized to meet the specific objectives of the downstream tasks. Furthermore, CIFT includes an automatic environment partitioner that segments the training data into a predefined number of virtual environments. This partitioning is achieved without relying on costly or privacy-sensitive domain annotation information, which is commonly required by many OoD generalization algorithms.

The technical details of CIFT are provided as follows.

**Parameter-efficient adaptors** CIFT incorporates three distinct parameter-efficient adapters, drawing inspiration from parameter-efficient fine-tuning strategies employed in large language models. These adapters enable the model to maintain its pretrained parameters $\theta$ fixed while introducing only a few adaptable, low-rank parameters $\Delta\theta$ into the existing architecture.

As shown in Figure 3 (a), our approach employs the following trainable modules: 1) **Bottleneck adapter** (Houlsby et al., 2019): A low-rank multilayer perceptron (MLP) with ReLU activation is integrated sequentially over the SwiGLU and attention modules. 2) **LoRA adapter** (Hu et al., 2021): It is implemented by inserting low-rank decomposition linear projection layers into the key-value projection layer of the attention modules parallelly. 3) **Layer Normalization**: The parameters of the normalization layers are made trainable to enhance adaptability during fine-tuning. Our empirical findings show that these adapters reduce the memory footprint and improve OoD performance compared to full-parameter tuning while offering greater adaptability than linear probings (Kumar et al., 2022; Alain & Bengio, 2018) for assessing large pre-trained models. The rank of bottleneck and LoRA adapters, denoted by $r$, is a tunable hyperparameter.

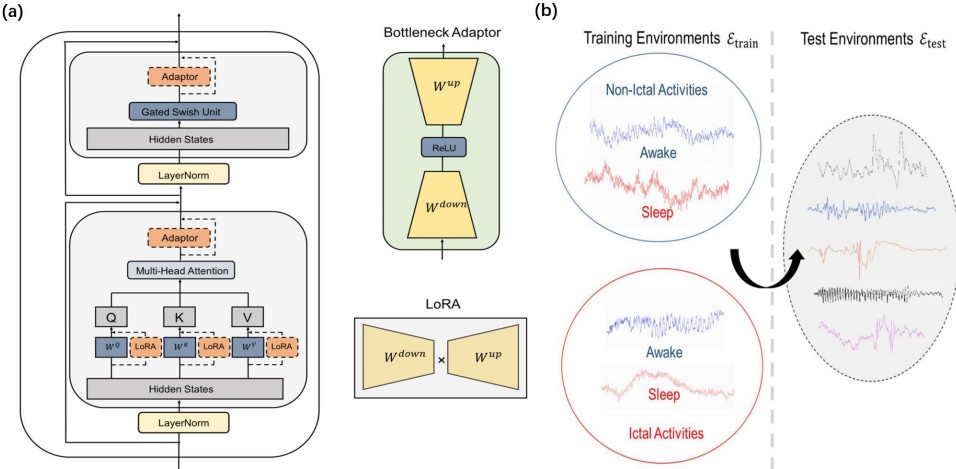

Figure 3: (a) Illustration of Trainable Modules During Fine-Tuning. Three types of trainable modules are inserted to pre-trained Transformer blocks to achieve low-resource generalization on downstream tasks. (b) Illustration of Out-of-Domain Generalization Tasks for EEG in Our Work.

**Contrastive fine-tuning loss** We adopt a dual approach to our loss functions, working together towards a common objective. For classification tasks, we implement Cross-Entropy (CE) loss, the standard fine-tuning target in previous studies. While this straightforward classification loss has yielded promising results through various fine-tuning strategies, we can further enhance it by addressing potential biases in the training environments. As illustrated in Figure 3 (b), EEG data is

highly heterogeneous; even samples from the same subject may not neatly fit into distinct domains conducive to effective domain adaptation or generalization.

Given a common trainable prompt $W = \{w_i, \ldots, w_L\} \in \mathbb{R}^{d \times L}$, where $d$ represents the model dimension and $L$ the prompt length, we introduce an automated prompt generation method to enhance the richness of EEG information: spectral and temporal prompt generation. In this method, spectral and temporal tokens are processed through a cross-attention module, yielding modal-specific prompts $U = \{u_i, \ldots, u_L\}$ and $V = \{v_i, \ldots, v_L\} \in \mathbb{R}^{n \times L}$. We append a special trainable token [CLS] to the sequence of each EEG channel at the initial position to capture global features for environment inference. The concatenated quadruple $\text{Concat}([\text{CLS}]; U, V; X)$ is processed as a whole by the network, where $X$ is the spectral tokens similar to those fed to the model in the pre-training stage, $\text{Concat}$ denotes concatenation operation. The prompts $U$ and $V$ are then projected into a $d$-dimensional latent space by two MLP projectors $g$ and $h$. For the temporal and spectral vectors in latent space, we apply CLIP loss for contrast. Thus, the overall loss function for our model is a combination of these two distinct losses, expressed as

$$\ell_{\text{CIFT}} = \lambda \cdot \ell_{\text{CLIP}} + (1 - \lambda) \cdot \ell_{\text{CE}}, \tag{5}$$

where $\lambda$ is a tunable hyperparameter. Unless otherwise stated, we set $\lambda = 0.1$

**Environment-aware reweighting** While CLIP loss enhances model performance by maximizing the boundaries between different samples and incorporating additional temporal information alongside spectral data, it does not necessarily lead to the learning of environment-invariant embeddings for decision-making. As depicted in Figure 4, we introduce an environment partitioner specified by an MLP-parameterized function $\rho$ with hyperparameter $K$. Here, $K$ denotes the number of virtual environments; in our context, determining exact environment labels for each sample is often costly or involves sensitive personal information. Consequently, $K$ is empirically set and serves as a surrogate rather than a representation of ground truth environment labels. We assume that each environment can be represented by a vector in a $K$-dimensional simplex $\Delta^K$, meaning each environment is a linear combination of $K$ basis environments. The environment labels are predicted by a function $\rho : \mathcal{X} \to \Delta^K$ parameterized by $\eta \in \mathbb{R}^D$. As we will demonstrate empirically, this approach yields comparable or superior results compared to methods that explicitly use environment labels for domain generalization.

Before calculating the training loss, we first aggregate the spectral and temporal features using a feature aggregation operator, which in this work is implemented as a simple global averaging:

$$\hat{U}, \hat{V} = \text{Aggregate}(U), \text{Aggregate}(V). \tag{6}$$

We assume that $\hat{U}$ and $\hat{V}$ contain environment-invariant attributes useful for classification, as well as environment-specific parts sensitive to environmental shifts. Our strategy prioritizes the environment-invariant features through a feature selection operator using masks generated by a differentiable Heaviside function (Otte, 2024):

$$m_U, m_V = \text{DifferentiableHeaviside}(\hat{U}), \quad \text{DifferentiableHeaviside}(\hat{V}),$$
$$\hat{U}_{\text{specific}}, \hat{V}_{\text{specific}} = \hat{U} \odot (1 - m_U), \quad \hat{V} \odot (1 - m_V) \tag{7}$$
$$\hat{e} = \rho(\text{Aggregate}(\text{Concat}\{\hat{U}_{\text{specific}}; \hat{V}_{\text{specific}}\}))$$

where $\hat{e}$ represents the estimated environment labels, $\odot$ is the Hadamard product. The target class labels are predicted by aggregating the invariant parts, which is achieved by addition, and fed to the classification head.

$$\ell_{\text{CE}} = y \log(\hat{y}) + (1 - y) \log(1 - \hat{y}), \quad \hat{y} = (\text{ClassificationHead}\{\hat{U}_{\text{invariant}} + \hat{V}_{\text{invariant}}\}),$$
$$\hat{U}_{\text{invariant}}, \hat{V}_{\text{invariant}} = \hat{U} \odot m_U, \quad \hat{V} \odot m_V \tag{8}$$

The contrastive objective is calculated as:

$$\ell_{\text{CLIP}} = -\log \frac{\exp\left(\langle \hat{U}_{\text{invariant},i}, \hat{V}_{\text{invariant},i} \rangle / \tau\right)}{\sum_{j=1}^{B} \exp\left(\langle \hat{U}_{\text{invariant},i}, \hat{V}_{\text{invariant},j} \rangle / \tau\right)} + \log \frac{\exp\left(\langle \hat{V}_{\text{invariant},i}, \hat{U}_{\text{invariant},i} \rangle / \tau\right)}{\sum_{j=1}^{B} \exp\left(\langle \hat{V}_{\text{invariant},i}, \hat{U}_{\text{invariant},j} \rangle / \tau\right)}, \tag{9}$$

where $B$ is the batch size. Finally, the CIFT loss (5) is calculated using (9) and (8), then reweighted using $\hat{e}$ for each sample with the batch, along with a Jacobian regularizer to transform objective (1) into a more feasible form, and we only have to minimize the surrogate loss functsion with regard to fine-tuning parameters $\Delta\theta$ as follows:

$$\mathcal{L}_{\text{CIFT}} = \mathbb{E}_{\mathbb{P}_{\mathbf{X}^{\hat{e}}, y^{\hat{e}}}} \ell_{\text{CIFT}}(f(x), y)\hat{e} + \beta \max_{\eta} \|\nabla_{\eta} \mathbb{E}_{\mathbb{P}_{\mathbf{X}^{\hat{e}}, y^{\hat{e}}}} \ell_{\text{CIFT}}(f(x), y)\hat{e}\|_2^2, \tag{10}$$

where $beta$ is a hyperparameter. Unless otherwise stated, we set $\beta = 10$.

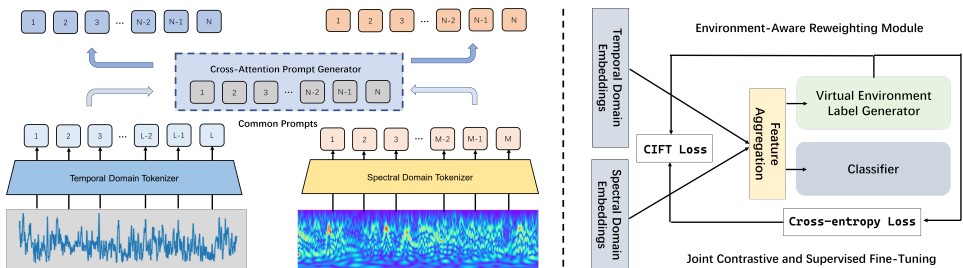

Figure 4: **Illustration of our proposed CIFT**. CIFT employs a dual-branch approach to generate spectral and temporal prompts, facilitating environment-aware learning for the learning of generalizable embeddings.

## 4 EXPERIMENTS

### 4.1 EXPERIMENTAL SETUP

**OoD baselines** We adopt the following baselines for comparison: VRM (Zhang et al., 2018), IRM (Arjovsky et al., 2019), V-REx (Krueger et al., 2021), IB-IRM (Ahuja et al., 2021), and EIIL (Creager et al., 2021), all of which, like our method, utilize Jacobian regularization for domain generalization. Additionally, we include a range of other baselines, such as Group DRO (Ghosal & Li, 2023), LearnMixin (Clark et al., 2019), CORAL (Sun et al., 2017), HEX (Wang et al., 2019), EnD (Ghaddar et al., 2021), DFA (Wang et al., 2024a), RUBI (Tian et al., 2022), and LfF (Nam et al., 2020). Unlike our approach, these methods explicitly use subject identities as environment labels to enhance domain generalization. We also incorporate contrastive algorithms known to benefit OoD generalization that, which can perform spectrotemporal alignment like CIFT, including InfoNCE (Harary et al., 2022), HSIC (Galstyan et al., 2022), SelfReg (Kim et al., 2021), and RELIC (Mitrovic et al., 2020)

**Evaluation metrics** To comprehensively evaluate the experimental results, we use precision, recall, F1- and F2-score as evaluation metrics. F2-score is adopted in critical applications that value information retrieval more than accuracy (i.e., accepting a relatively large number of false positives but virtually guaranteeing that all the true positives are found). In the biomedical scenario, F2-score is more valued than F1-score, since ignoring any seizure is costly in diagnosis. Other metrics, including accuracy, AUCROC and AUCPR, are also reported. For tasks involving non-epileptic subjects, we employ metrics that are most sensitive to performance nuances and align with established research practices, which are detailed in their respective sections.

### 4.2 MAIN RESULTS

In this section, we present our primary findings on the challenging tasks of Out-of-Distribution (OoD) Seizure Detection (SD) and Seizure Forecasting (SF). Our focus transcends the traditional assessment of an algorithm's average performance, as we seek to evaluate CIFT's capacity to generalize across diverse disease subtypes and institutes. This approach is critical for practical epilepsy monitoring applications. Typically, neurologists have limited prior knowledge of a patient's pathology until a sufficient number of seizures have been clinically validated. Thus, a model's ability to generalize effectively over heterogeneous patient profiles and data sources is of paramount importance for its real-world utility in seizure prediction and detection.

**Datasets** For SD, we evaluate our approach on a self-collected dataset featuring recordings from patients with clonic seizures for training and atonic seizures for testing, providing a benchmark for out-of-disease generalization. Details are provided in Appendix E.4. For SF, we utilize a benchmark constructed from intracranial EEG recorded in four hospitals (Li et al., 2023). Its heterogeneity stems from variations in epileptogenic lesions and recording conditions, further complicates forecasting tasks and serves as a benchmark for out-of-institute generalization. Details are provided in Appendix E.3. To ensure fair comparison, all baselines are equipped with identical PEFT adapters used in this study during fine-tuning, except for ERM-Full, ERM-LP, and ERM-LoRA, which denote full-parameter, linear probing, and LoRA baselines, respectively. The rank of adapters $r = 16$, and the number of virtual environments $K = 8$. Results are averaged over three runs of random seeds.

| Method | Category | Acc. | | F1 | | F2 | | AUCROC | | AUPRC | |
|---|---|---|---|---|---|---|---|---|---|---|---|
| | | SD | SF | SD | SF | SD | SF | SD | SF | SD | SF |
| ERM-LoRA | SF | 0.688 | 0.674 | 0.489 | 0.310 | 0.476 | 0.260 | 0.692 | 0.681 | 0.614 | 0.252 |
| ERM | SF | 0.799 | 0.719 | 0.416 | 0.280 | 0.449 | 0.254 | 0.859 | 0.673 | 0.654 | 0.245 |
| ERM-LP | SF | 0.750 | 0.527 | 0.548 | 0.364 | 0.605 | 0.272 | 0.702 | 0.719 | 0.654 | 0.339 |
| ERM-Full | SF | 0.833 | 0.644 | 0.574 | 0.331 | 0.658 | 0.267 | 0.776 | 0.685 | 0.699 | 0.269 |
| VRM | VR | 0.672 | 0.810 | 0.472 | 0.267 | 0.681 | 0.374 | 0.662 | 0.809 | 0.470 | 0.367 |
| IRM | INVR | 0.709 | 0.729 | 0.595 | 0.086 | 0.676 | 0.092 | 0.790 | 0.490 | 0.733 | 0.145 |
| V-REx | INVR | 0.708 | 0.733 | 0.612 | 0.103 | 0.634 | 0.145 | 0.794 | 0.621 | 0.739 | 0.197 |
| IB-IRM | INVR | 0.806 | 0.707 | 0.623 | 0.128 | 0.588 | 0.109 | 0.829 | 0.635 | 0.748 | 0.194 |
| EIIL | INVR | 0.853 | 0.686 | 0.760 | 0.146 | 0.609 | 0.119 | 0.903 | 0.636 | 0.878 | 0.207 |
| Group DRO | RO | 0.827 | 0.724 | 0.578 | 0.235 | 0.712 | 0.229 | 0.832 | 0.686 | 0.790 | 0.228 |
| LearnMixin | DA | 0.695 | 0.738 | 0.020 | 0.282 | 0.400 | 0.292 | 0.877 | 0.689 | 0.843 | 0.235 |
| CORAL | DA | 0.854 | 0.645 | 0.714 | 0.322 | 0.689 | 0.331 | 0.873 | 0.680 | 0.838 | 0.233 |
| HEX | FD | 0.807 | 0.711 | 0.542 | 0.028 | 0.831 | 0.049 | 0.864 | 0.368 | 0.837 | 0.121 |
| EnD | FD | 0.717 | 0.768 | 0.662 | 0.457 | 0.667 | 0.461 | 0.833 | 0.797 | 0.786 | 0.351 |
| DFA | FD | 0.866 | 0.717 | 0.668 | 0.488 | 0.781 | 0.487 | 0.865 | 0.840 | 0.803 | 0.414 |
| RUBI | FD | 0.864 | 0.173 | 0.750 | 0.280 | 0.652 | 0.085 | 0.887 | 0.782 | 0.854 | 0.366 |
| LfF | FD | 0.876 | 0.788* | 0.784* | 0.540 | 0.712 | 0.521* | 0.903* | 0.875* | 0.876* | 0.595* |
| InfoNCE | CL | 0.869 | 0.662 | 0.727 | 0.361 | 0.692* | 0.364 | 0.894 | 0.701 | 0.850 | 0.428 |
| HSIC | CL+FD | 0.859 | 0.756 | 0.722 | 0.386 | 0.640 | 0.401 | 0.892 | 0.835 | 0.834 | 0.441 |
| SelfReg | CL+DA | 0.436 | 0.321 | 0.350 | 0.322 | 0.322 | 0.371 | 0.516 | 0.590 | 0.423 | 0.173 |
| RELIC | CL+INVR | 0.874* | 0.865 | 0.796 | 0.520 | 0.588 | 0.605 | 0.909 | 0.936 | **0.879** | 0.769 |
| Beatrix + CIFT | CL+INVR | **0.938** | **0.918** | **0.901** | **0.753** | **0.901** | **0.742** | **0.988** | **0.948** | **0.975** | **0.832** |

Table 1: **Comparison of CIFT with Other Methods on Seizure Detection (SD) and Seizure Forecasting (SF) Tasks Across Key Metrics.** Methods that explicitly leverage environment partitioning of training data, requiring annotations from domain experts and/or subject identity information, are highlighted in light green. In contrast, methods that do not rely on explicit ground truth environment labels are marked in dark green. Those fine-tuned naïvely with ERM but using distinct parameter configurations are denoted in light blue. We categorize the algorithms based on their underlying mechanisms as follows: VR: Vicinal Representation; INVR: Invariant Representation; RO: Robust Optimization; DA: Domain Alignment; FD: Feature Disentanglement; CL: Contrastive Learning. We mark metric values ranking the **first**, the second and the third*.

**CIFT improves OoD generalization among epileptic subjects** As demonstrated in Table 1, our approach surpasses alternative domain generalization methods. Beatrix, when fine-tuned in the spectral domain, showed comparable OoD performance across full-parameter, linear probing, and LoRA methods. However, low-rank fine-tuning excelled in OoD generalization with reduced memory requirements. Additional OoD techniques led to incremental performance gains.

Remarkably, incorporation of temporal prompts via tokenization and contrastive alignment has demonstrated consistently better performance across non-contrastive baselines with the exception of SelfReg, which, despite its optimization for image data, exhibits less effectiveness on EEG data due to its structural prior. More significantly, our approach outperforms InfoNCE, a baseline lacking CIFT's environment predictor and gradient regularization. This highlights the significant role of environment-aware design, in conjunction with spectrotemporal information integration, in enhancing the model's effectiveness. Besides, RELIC achieves the second-best performance overall but did not surpass CIFT. This can be attributed to the implicit treatment of each distinct sample as a separate environment when enforcing domain invariance (Mitrovic et al., 2020), whereas ours partitions the training data into a manageable number of environments.

**Can CIFT be extended to other architectures?** We investigate the flexibility of CIFT by applying it to other architectures that operate on the time-frequency representation of EEG, assessing whether CIFT can enhance the performance of spectral EFMs despite inherent discrepancies in pre-trained models due to differing data and architectural specifications. We utilize BrainBERT (Wang et al., 2023) and ScatterFormer (Zheng et al., 2023), both of which are Transformer-based models trained on extensive epilepsy datasets. Table 2 demonstrates CIFT's significant performance enhancements. Strikingly, Beatrix outperforms BrainBERT in seizure forecasting despite BrainBERT's specialization in intracranial recordings and Beatrix's lack of such data in pre-training. This underscores Beatrix's superior representations, which CIFT further bolsters. BrainBERT's isolated electrode processing misses out on the interchannel dynamics present in Beatrix, and ScatterFormer's fixed electrode requirement hinders its versatility with variable intracranial EEG setups. Our approach is able to overcome these deficiencies in a multi-faceted manner.

| Method | Acc. | | F1 | | F2 | | AUC-ROC | | AUPRC | |
|---|---|---|---|---|---|---|---|---|---|---|
| | SD | SF | SD | SF | SD | SF | SD | SF | SD | SF |
| ScatterFormer-RELIC | 0.771 | 0.788 | 0.540 | 0.424 | 0.582 | 0.461 | 0.818 | 0.781 | 0.473 | 0.465 |
| ScatterFormer-CIFT | 0.875 | 0.807 | 0.644 | 0.508 | 0.778 | 0.512 | 0.763 | 0.830 | 0.752 | 0.513 |
| BrainBERT-RELIC | 0.750 | 0.668 | 0.792 | 0.367 | 0.906 | 0.364 | 0.960 | 0.703 | 0.813 | 0.314 |
| BrainBERT-CIFT | 0.792 | 0.793 | 0.890 | 0.402 | 0.910 | 0.460 | 0.981 | 0.814 | 0.963 | 0.460 |
| Beatrix + RELIC | 0.874 | 0.865 | 0.796 | 0.520 | 0.588 | 0.605 | 0.909 | 0.936 | 0.879 | 0.769 |
| Beatrix + CIFT | 0.938 | 0.918 | 0.901 | 0.753 | 0.901 | 0.742 | 0.988 | 0.948 | 0.975 | 0.832 |

Table 2: **OoD Performance comparison of CIFT in different spectrally pre-Trained EEG Models** This table demonstrates how CIFT enhances the performance across various EEG models initially pre-trained on spectrogram representation of brain signals. It is observed that CIFT consistently improves performance in terms of multiple metrics, especially F1, F2 and AUCROC, despite variations in pre-training data and architectural designs.

**Ablation study and hyperparameter analysis** We perform an extensive ablation study along with hyperparameter analysis of CIFT. The outcomes of these experiments are detailed in Appendix A.2 and A.3. Additional experiments on the self-collected dataset as well as another two open datasets can be found in Appendix A.1

## 4.3 FURTHER EXTENSIONS

Building upon our preliminary findings, we explore the potential of the proposed approach by conducting more experiments on OoD tasks involving EEG records from non-epileptic subjects.

**Auditory Brain Decoding** Evaluating our approach on a public EEG audio decoding benchmark (Broderick et al., 2018) in alignment with the experimental protocols established by Défossez et al. (2023). Notably, our results, as delineated in Table 8, Appendix A.4, corroborate the capability of the CIFT-tuned Beatrix model to amplifies the model's efficacy in deciphering neuroelectrophysiological responses to natural speech.

**Motor Imagery** Motor imagery classification, which involves identifying brain activity associated with mentally simulated movements, is a pivotal area of research with substantial implications for developing rehabilitation strategies and assistive technologies. We experimented on a PhysioNet motor imagery benchmark (Schalk et al., 2004) used in previous works such as (Yuan et al., 2024a) As delineated in Table 9, Appendix A.5, our approach demonstrates superiority over not only other spectral EEG models, such as BrainBERT and TSFF-Net (Miao & Zhao, 2024), a fully supervised baseline but also temporal EFMs like Brant and MBrain that has been proven achieving strong results for BCI tasks, underscoring the model's proficiency in discerning event-related potentials across individuals.

**Sleep Staging** Sleep staging serves as a fundamental benchmark in EEG analysis for assessing a model's OoD generalizability in multi-class scenario. Our approach surpasses various EEG foundation models and even some fully-supervised models that have been specifically tailored for sleep monitoring. More details of sleep staging are in Appendix A.6.

## 5 CONCLUSION AND FUTURE WORK

In this paper, we introduce Beatrix, a pioneering EEG foundation model that demonstrates superior out-of-distribution (OoD) generalization over current state-of-the-art (SOTA) models for a range of seizure-related and non-seizure tasks, all with a significantly reduced fine-tuning cost. To bolster Beatrix's OoD generalization capabilities, we have developed CIFT, a novel contrastive invariance learning technique. CIFT delivers substantial performance improvements by effectively inferring environmental contexts and seamlessly integrating spectrotemporal information.

While Beatrix is currently optimized for domain-specific applications, such as epilepsy monitoring and forecasting, it has yielded promising results for both healthy individuals and those with non-epileptic neurological conditions. Looking ahead, our future research will be directed towards expanding the versatility of our approach. We aim to embrace an even broader array of downstream tasks and to incorporate a variety of brain modalities, including MEG and fMRI, into a unified, comprehensive generative multimodal foundation model. This advancement will not only amplify its predictability and generalizability in real-world clinical settings.

## 6 ETHICS STATEMENT

The public datasets utilized in this study are freely accessible and designated for academic research purposes and are not associated with any privacy or security concerns. We adhere to the ethical guidelines for data usage with meticulous attention to each dataset's specific requirements. Regarding the private data incorporated in our research, stringent measures were taken to ensure anonymity and data desensitization. The use of such data was granted by the hospital's ethics committee after a thorough review process. Furthermore, we obtained explicit informed consent from all participants, ensuring they agree that their data would be employed and shared for academic purposes.

It is crucial to emphasize that the results of this study are purely for scientific exploration and have not been subjected to clinical validation. Consequently, they should not be interpreted as support for any clinical advice or practice.

## 7 REPRODUCIBILITY STATEMENT

To bolster the reproducibility of our research and to pave the way for future studies on EEG foundation models, we have meticulously compiled an extensive collection of open EEG datasets, which we believe to be the most comprehensive to date. These datasets are publicly accessible, and we have detailed their characteristics along with download links in Appendix **??**. Additionally, we have included a comprehensive description of the data cleansing and preprocessing steps employed in this study. The source code is included in the supplementary materials. For the convenience of reproducing the experimental outcomes, the preprocessed private benchmark dataset can be accessed via the following anonymous link: https://drive.google.com/drive/folders/1eLzx_FrfLjZLs3cnkATsRUrkaU-L0HPd?usp=sharing. Upon publication of the paper, the raw EEG recordings from the subjects in the private benchmark will also be released. This will support the advancement of out-of-distribution (OoD) generalizable EEG applications and contribute to the progress of clinical research of seizure-related neurological disorders.

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

# A   More Experimental Results

## A.1   Additional Experiments Among Epileptic Subjects

We expand upon our primary experiments by benchmarking Beatrix against a range of pre-trained EEG models on more epileptic subjects.

| Dataset Model | CHB-MIT | | | | Siena | | | |
|---|---|---|---|---|---|---|---|---|
| | Modality | Pre. | Rec. | F1 | F2 | Modality | Pre. | Rec. | F1 | F2 |
| TF-C | TD + FD | 0.178 | 0.614 | 0.180 | 0.276 | TD + FD | 0.080 | 0.616 | 0.137 | 0.249 |
| SimMTM | TD | 0.543 | 0.368 | 0.428 | 0.388 | TD | 0.323 | 0.434 | 0.275 | 0.330 |
| One Fits All | TD | 0.511 | 0.566 | 0.512 | 0.537 | TD | 0.473 | 0.437 | 0.430 | 0.430 |
| MBrain | TD | 0.534 | 0.529 | 0.494 | 0.504 | TD | 0.389 | 0.610 | 0.459 | 0.533 |
| LaBraM | TD | 0.410 | 0.570 | 0.529 | 0.477 | TD | 0.318 | 0.680 | 0.433 | 0.548 |
| BIOT | TD | 0.177 | 0.277 | 0.205 | 0.216 | TD | 0.269 | 0.353 | 0.305 | 0.173 |
| Brant | TD | 0.574 | 0.614 | 0.580 | 0.597 | TD | 0.431 | 0.660 | 0.484 | 0.553 |
| Brant-2 | TD + FD | 0.538 | 0.670 | 0.595 | 0.637 | TD + FD | 0.499 | 0.700 | 0.566 | 0.634 |
| BrainBERT | SD | 0.525 | 0.594 | 0.551 | 0.574 | SD | 0.473 | 0.606 | 0.486 | 0.535 |
| ScatterFormer | SD | 0.557 | 0.649 | 0.598 | 0.627 | SD | 0.503 | 0.675 | 0.535 | 0.590 |
| Beatrix + CIFT | SD | **0.603** | **0.752** | **0.652** | 0.703 | SD | 0.574 | 0.750 | 0.649 | 0.705 |

Table 3: **OoD Performance Comparison of different models on Two Public EEG Seizure Detection Benchmarks**. Models that are fine-tuned from publicly available checkpoints are highlighted in green, and models that cannot be reproduced for fine-tuning due to lack of publicly available implementations and/or pre-trained checkpoints are highlighted in blue, for which the metrics were reported using the results reported in their respective works. TD: Temporal Domain. FD: Fourier Domain. SD: Spectral Domain. Pre.: Precision. Rec.: Recall. Model names highlighted in blue represent outcomes reported in their original publications, whereas those highlighted in green are outcomes reproduced by our own work. We mark metric values ranking the **first**, the second and the third*.

| Seizure SubType | Model | Modality | F1 | F2 | AUCROC | AUCPR |
|---|---|---|---|---|---|---|
| **Absence Seizure** | LaBraM | TD | 0.515 | 0.631 | 0.473 | 0.430 |
| | BIOT | TD | 0.514 | 0.603 | 0.499 | 0.542 |
| | Beatrix + CIFT | SD + TD | **0.607** | **0.648** | **0.774** | **0.652** |
| **Atonic Seizure** | LaBraM | TD | 0.465 | 0.525 | 0.561 | 0.893 |
| | BIOT | TD | 0.558 | 0.648 | 0.674 | 0.911 |
| | Beatrix + CIFT | SD + TD | **0.772** | **0.733** | **0.967** | **0.848** |
| **Clonic Seizure** | LaBraM | TD | 0.491 | 0.590 | 0.721 | 0.538 |
| | BIOT | TD | 0.399 | 0.440 | 0.542 | 0.354 |
| | Beatrix + CIFT | SD + TD | **0.788** | **0.750** | **0.868** | **0.810** |

Table 4: **OoD performance comparison of different models on the self-collected dataset in out-of-subject scenario**. CIFT significantly improves out-of-subject generalization on various seizure subtypes. TD: Temporal Domain; SD: Spectral Domain.

### A.1.1   More experiments on the self-collected dataset

We have demonstrated our approach's effectiveness in an out-of-disease case in the main results. In Table 4, we further present outcomes in an out-of-subject scenario, further illustrating that CIFT enhances out-of-subject performance. The results are averaged over 5 randomly split folds, ensuring that no subjects overlap between folds.

### A.1.2   More experiments on CHB-MIT and Siena datasets

We evaluate our approach using two scalp EEG datasets in an out-of-subject setting to assess its out-of-distribution (OoD) generalizability across different subjects. Additionally, we benchmark our model against other state-of-the-art EEG models to demonstrate its effectiveness in capturing variability across diverse populations.

**Experimental setup** We describe the datasets and experimental settings as follows.

The CHB-MIT dataset (Shoeb, 2009) comprises 23-channel EEG recordings captured at a sampling rate of 256 Hz from a cohort of 22 subjects diagnosed with intractable epilepsy. These subjects exhibit a wide variety of seizure types, adding to the complexity and diversity of the dataset.

The Siena dataset (Detti et al., 2020), conversely, consists of 27-channel EEG recordings from 14 patients, with a higher sampling rate of 512 Hz, providing a denser temporal resolution of brain activity.

For a comprehensive comparison, we not only focus on spectral domain models such as BrainBert and ScatterFormer but also include several time-domain EEG feature models (EFMs), including MBrain, Brant and Brant-2. These models are noted for their significantly larger parameter counts during both pre-training and fine-tuning stages, offering a stark contrast to our approach.

Our performance metrics, the F1 and F2 scores, are derived from the average of five cross-validation folds, ensuring that each fold contains distinct subjects. To ensure fair comparison, we conduct the experiments following setups used in previous work such as Yuan et al. (2024a;b); Yang et al. (2023). This rigorous cross-validation strategy eliminates any bias that might arise from subject overlap across folds. However, due to some previous works not providing their code or model checkpoints for reproduction—often due to intellectual property concerns or the reliance on private datasets—some metrics mentioned in Section 4.2 could not be directly compared. As a result, we report the maximum overlap of available metrics from the various studies for a fair comparison. The rank of adaptors $r = 16$, and the number of virtual environments $K = 8$.

**Results** As shown in Table 3, our approach demonstrates a significant improvement in out-of-distribution (OoD) generalization on both the CHB-MIT and Siena datasets. Notably, our model outperforms large-scale time-domain EFMs in both F1 and F2 scores, highlighting its ability to capture the subtle complexities of EEG data across diverse subject populations and recording conditions. These results further reinforce the OoD generalizability of our model.

Interestingly, we observed that other spectral domain models also exhibit impressive performance compared to time-domain models such as LaBraM and BIOT, with our approach trailing only behind Brant and Brant-2. These latter models, however, benefit from a significantly larger parameter count and incorporate Fourier domain features to complement their time-domain representations. This finding underscores the critical role of spectral representation in effectively integrating both time and frequency domain information for enhancing EEG analysis.

## A.2 ABLATION STUDY

**Do Mother Wavelets Affect Beatrix's OoD Performance?** We investigate the effect of mother wavelet choice on model performance during testing. As Table 5 shows, the impact is minimal. Since our pre-training protocol randomly selects mother wavelet functions for each training iteration, the variability in time-frequency representations, generated by a diverse set of analytic mother wavelets, appears to inoculate the model against overfitting to specific patterns associated with any single wavelet. By exposing the model to a wide array of wavelet functions, we effectually desensitize the model against these subtleties, thereby maintaining its robustness during the fine-tuning phase.

| Configuration | Acc. | | F1 | | F2 | | AUCROC | | AUPRC | |
|---|---|---|---|---|---|---|---|---|---|---|
| | SD | SF | SD | SF | SD | SF | SD | SF | SD | SF |
| CMH | 0.865 | 0.911 | 0.659 | 0.697 | 0.623 | 0.733 | 0.952 | 0.954 | 0.845 | 0.839 |
| GMW($\beta = 16, \gamma = 1$) | 0.917 | 0.915 | 0.791 | 0.860 | 0.813 | 0.910 | 0.960 | 0.954 | 0.906 | 0.936 |

Table 5: **Influence of mother wavelets on seizure detection and forecasting tasks.** Performance discrepancies caused by different mother wavelet at feature extraction stage is minor. CMH: Complex Mexican Hat. GMW: Generalized Morse Wavelet.

**Do Low-Rank Adaptors Affect Beatrix's OoD Performance?** We assess the impact of various low-rank adaptors within the CIFT framework on Beatrix's out-of-distribution (OoD) performance. Experimental results in Table 6, indicate that the removal of any adaptor module leads to decline in performance, albeit to varying degrees. The negative effect resulted from removal of the normalization module is minimal. This may stem from the compensatory capabilities of the bottleneck and

LoRA modules, which can counteract the absence of adjustable parameters of layer normalization to some extent.

| Configuration | Acc. | | F1 | | F2 | | AUCROC | | AUPRC | |
|---|---|---|---|---|---|---|---|---|---|---|
| | SD | SF | SD | SF | SD | SF | SD | SF | SD | SF |
| CIFT w/o Bottleneck | 0.8292 | 0.8639 | 0.5773 | 0.6755 | 0.5738 | 0.5930 | 0.8778 | 0.9476 | 0.7004 | 0.7160 |
| CIFT w/o LoRA | 0.7542 | 0.8444 | 0.2716 | 0.4286 | 0.4167 | 0.4817 | 0.8052 | 0.9124 | 0.5600 | 0.6371 |
| CIFT w/o Norm | 0.8458 | 0.8806 | 0.7517 | 0.7034 | 0.6780 | 0.6281 | 0.9301 | 0.9569 | 0.8447 | 0.7961 |

Table 6: **Influence of low-Rank adaptor configurations on seizure detection and forecasting tasks.** The table illustrates the substantial decline in performance following the removal of the LoRA module, while the deletion of layer normalization has the least detrimental effect.

## A.3 HYPERPARAMETER ANALYSIS

In this section, we delve into the sensitivity analysis of two pivotal hyperparameters within the Contrastive Invariant Fine-Tuning (CIFT) method: the rank $r$ and the count of virtual environments $K$. The rank indictates the dimensionality of the low-rank adaptation, while $K$ influences the granularity of environmental distinctions. During our analysis of $r$, we fixed $K$ at 8, and for the examination of $K$, we set $r$ to 16.

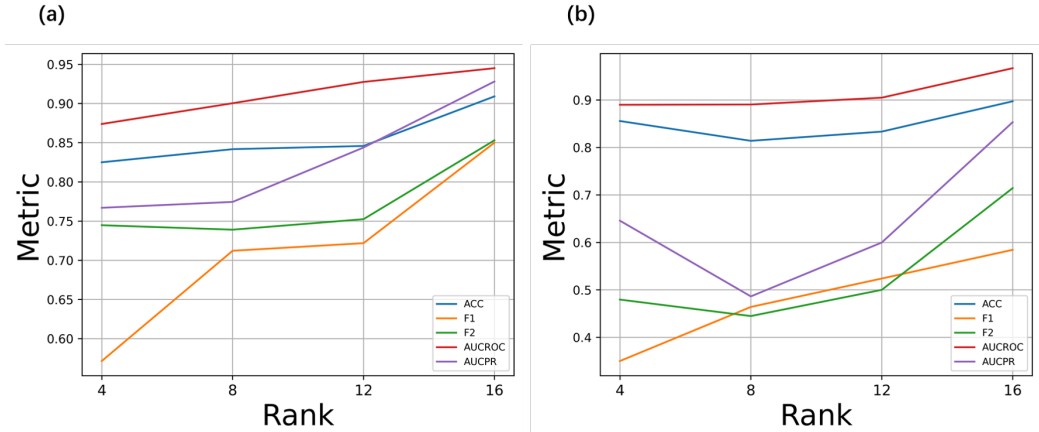

Figure 5: **Hyperparameter analysis of rank $r$.** (a) Seizure detection. (b) Seizure forecasting.

Results in Figure 5 reveal a positive correlation r and the model's performance. Notably, there is a progressive enhancement in performance with an increase in $r$, yet the rate of improvement tapers off as $r$ grows larger. This suggests a diminishing return on increasing the rank, indicating an optimal range for $r$ beyond which the gains in performance are marginal. Still, a higher rank allows the model to capture more complex patterns, leading to better performance.

| Configuration | Acc. | | F1 | | F2 | | AUC-ROC | | AUPRC | |
|---|---|---|---|---|---|---|---|---|---|---|
| | SD | SF | SD | SF | SD | SF | SD | SF | SD | SF |
| CIFT $K = 4$ | 0.823 | 0.864 | 0.731 | 0.676 | 0.669 | 0.593 | 0.869 | 0.948 | 0.674 | 0.716 |
| CIFT $K = 8$ | 0.917 | 0.915 | 0.791 | 0.860 | 0.813 | 0.910 | 0.960 | 0.954 | 0.906 | 0.936 |

Table 7: **Hyperparameter analysis of number of virtual environments $K$.**

The performance of the Contrastive Invariant Fine-Tuning (CIFT) method is also influenced by the number of virtual environments $K$, but not as significant as $r$. Performance metrics reported in Table 7 and $t$-SNE manifold analysis of learned domain-invariant embeddings shown in Figure 6 suggest that an increased $K$ can be beneficial for learning environment-invariant features, which results in less scattered distribution of unseen data at test time by forcing the model to discard more fine-grained domain-specific features during fine-tuning time.

Given that CIFT is designed to automatically identify environments and reweight the loss to prevent the acquisition of domain-specific features, we recommend tuning $K$ based on the underlying domains of the development dataset. In practical applications, $K$ should be set higher than the potential number of domains, taking into account real-world variables like patient outcomes and MRI structural features. However, caution should be exercised not to set $K$ excessively high, as an overly granular partitioning of environments may be susceptible to the stochastic fluctuations in brain dynamics. This could not only increase the computational burden during fine-tuning but also potentially compromise the model's ability to generalize across unseen data.

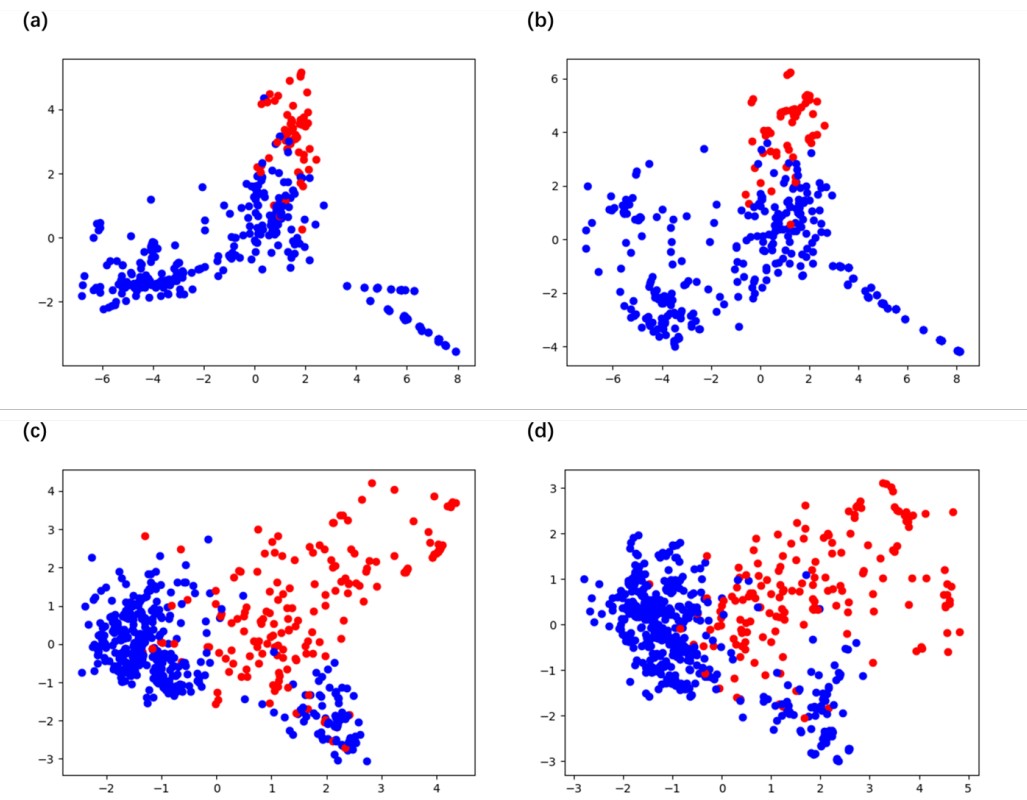

Figure 6: $t$-**SNE analysis of domain-invariant feature embedding used for classification with different $K$s at test time.** (a) Seizure forecasting ($K = 4$). (b) Seizure forecasting ($K = 8$). (c) Seizure detection ($K = 4$). (d) Seizure detection ($K = 8$).

### A.4 AUDITORY BRAIN DECODING

The realm of auditory brain decoding presents a significant challenge for EEG analysis, requiring models to accurately interpret complex neural signals associated with hearing.

**Experimental Setup** We use the EEG data recorded from English-speaking participants listened to extracts of *The Old Man and the Sea* by (Broderick et al., 2018). To ensure fair comparison, we follow settings in Défossez et al. (2023).

| Method | Random model | CNN | CLIP | Deep Mel | Wav2vec | Beatrix + CIFT |
|--------|--------------|-----|------|----------|---------|----------------|
| **Acc.** | 0.005 | 0.010 | 0.020 | 0.154 | 0.177 | **0.501** |

Table 8: **Performance in auditory EEG neural decoding**.

## A.5 MOTOR IMAGERY CLASSIFICATION

Detailed experimental results are shown in 9.

| Metrics Models | Motor Imagery | |
|---|---|---|
| | Acc. | F1 |
| TSFF-Net | 73.00 | 73.87 |
| TF-C | 60.06 | 57.79 |
| SimMTM | 57.48 | 57.37 |
| One Fits All | 71.25 | 72.56* |
| BrainBERT | 64.84 | 70.32 |
| MBrain | 61.06 | 60.42 |
| Brant | 72.00* | 71.84 |
| Beatrix + CIFT | **74.93** | **80.42** |

Table 9: **Performance on motor imagery classification.** Our approach brings significant gains in F1-score in comparison to both finetuned EEG foundation models and fully supervised models specially adapted for motor imagery tasks. Model names highlighted in blue represent outcomes reported in their original publications, whereas those highlighted in green are outcomes reproduced by our own work. We mark metric values ranking the **first**, the second and the third*.

## A.6 SLEEP STAGING

In sleep health research, sleep staging is essential for deepening our understanding of sleep states and patterns, aiding in the prevention and diagnosis of sleep-related disorders. We adopt definition in American Academy of Sleep Medicine (AASM) manual where sleep is divided into five stages: wake, N1, N2, N3, and REM. Thus, sleep stage classification is framed as a 5-class problem.

**Experimental setup** To evaluate model performance in sleep stage classification, we utilize two EEG datasets: SleepEDF and the Haaglanden Medisch Centrum (HMC) sleep staging database.

For SleepEDF, we use the SleepEDF-78 dataset, which consists of 153 whole-night polysomnographic recordings from 78 subjects during sleep cassette studies. The EEG data, sampled at 100Hz, includes one EEG channel, with recordings segmented into 30-second epochs. Subjects are randomly divided into five groups.

The HMC dataset contains 151 whole-night polysomnographic (PSG) recordings from 151 subjects, sampled at 256Hz across four EEG channels. Similar to SleepEDF, subjects are split into five groups, and the EEG signals are segmented into 30-second epochs.

In our comparative analysis, include several robust fully-supervised sleep staging models that have been specifically tailored for this task, including sDreamer (Chen et al., 2023), Eognet (Fan et al., 2021). For sepctral domain models, given that ScatterFormer is specialized for epilepsy-related tasks, we've chosen BrainBERT as the sole spectral domain model for our baseline comparison.Additionally, we benchmark against pre-trained EEG models operating in the time-domain, namely TF-C, Sim-MTM, One Fits All, MBrain, and Brant.

For the hyperparameters of CIFT, the rank of adaptors is set $r = 16$, and the number of virtual environments $K = 16$.

**Results** As demonstrated in Tables 10 and 11, our model not only achieves competitive performance but also surpasses existing benchmarks in critical metrics such as F1-score and Cohen's $\kappa$. These metrics are particularly significant as they directly reflect the model's ability to accurately classify sleep stages, a complex multi-class task fraught with inter- and intra-subject variability. In particular,

The consistent superiority of our model across diverse datasets, including the SleepEDFx and HMS Sleep Staging Benchmarks, underscores its robustness. This robustness is a testament to the model's sophisticated handling of the nuanced patterns present in EEG data during sleep staging. Furthermore, the model's broader applicability is underscored by its ability to enhance out-of-subject performance, a critical factor in real-world clinical applicability.

As shown in Table 10 and 11, our model achieves competitive or better performance in terms of F1-score and Cohen's $\kappa$. The outcomes further validate the broader applicability of our model to enhance out-of-subject performance in terms of multi-class tasks.

| Model | Modality | Rec. | Spe. | F1 | Cohen's $\kappa$ |
|---|---|---|---|---|---|
| sDreamer | TD | - | - | 0.705† | - |
| Eognet | TD | - | - | 0.693*† | - |
| TF-C | TD + FD | 0.514 | 0.902 | 0.494 | 0.507 |
| Sim-MTM | TD | 0.363 | 0.890 | 0.494 | 0.507 |
| One Fits All | TD | 0.563 | 0.912 | 0.548 | 0.550 |
| MBrain | TD | 0.584* | 0.922 | 0.582 | 0.598 |
| Brant | TD | 0.583 | 0.916 | 0.568 | 0.565 |
| BrainBERt | SD | 0.594 | 0.918* | 0.587 | 0.571 |
| Beatrix + CIFT | SD + TD | **0.788** | **0.958** | **0.788** | **0.721** |

Table 10: Out-of-Subject Generalization on SleepEDFx Dataset. CIFT consistently improves out-of-subject generalization for all multi-class metrics. TD: Temporal Domain. FD: Fourier Domain. SD: Spectral Domain. † Performance of fully-supervised models reported in previous work. Model names highlighted in blue represent outcomes reported in their original publications, whereas those highlighted in green are outcomes reproduced by our own work. We mark metric values ranking the **first**, the second and the third*.

| Model | Modality | Rec. | Spe. | F1 | Cohen's $\kappa$ |
|---|---|---|---|---|---|
| sDreamer | TD | - | - | 0.688*† | - |
| Eognet | TD | - | - | 0.719† | - |
| TF-C | TD + FD | 0.353* | 0.843 | 0.302 | 0.239 |
| Sim-MTM | TD | 0.315 | 0.832 | 0.273 | 0.177 |
| One Fits All | TD | 0.511 | 0.884 | 0.505 | 0.435 |
| MBrain | TD | 0.540 | 0.895 | 0.515 | 0.487 |
| Brant | TD | 0.419 | 0.859 | 0.381 | 0.304 |
| BrainBERT | SD | 0.531 | 0.890 | 0.520 | 0.465* |
| Beatrix + CIFT | SD + TD | **0.736** | **0.941** | **0.736** | **0.637** |

Table 11: **OoD performance comparison of different models on HMC dataset**. CIFT consistently improves out-of-subject generalization for all multi-class metrics. TD: Temporal Domain. FD: Fourier Domain. SD: Spectral Domain. † Performance of fully-supervised models reported in previous work. Model names highlighted in blue represent outcomes reported in their original publications, whereas those highlighted in green are outcomes reproduced by our own work.

## B DETAILS OF BASELINES

### B.1 MODEL BASELINES

In this section, we elaborate the model detailed introductions to the baselines used in our experiments.

- TF-C (Zhang et al., 2022). The time-frequency consistency model uses EEG features from both time and frequency domains for training. It employs contrastive learning with augmentations to ensure the consistency of embeddings across these domains. The model uses a novel consistency loss to align time-based and frequency-based representations effectively. It is trained on diverse datasets including EEG, EMG, and ECG signals.
- SimMTM (Dong et al., 2024). Simple Masked Time-series Modeling (SimMTM) leverages EEG features from time series data for pre-training. It utilizes a unique masked modeling approach by masking parts of the time series and training the model to reconstruct the original series from multiple masked versions. This method incorporates contrastive learning and masked modeling techniques. SimMTM is trained on a substantial amount of unlabeled data to improve the representations for downstream tasks like forecasting and classification.

- One Fits All (Zhou et al., 2023). This model uses EEG features represented through embeddings obtained from pre-trained language models like GPT-2. It employs a contrastive learning approach and masked modeling techniques to fine-tune the model for various time series analysis tasks, including classification, anomaly detection, and forecasting. This method leverages extensive pre-training on large datasets, typically exceeding 10GB, to ensure robust performance across different applications.

- BrainBERT (Wang et al., 2023). BrainBERT is a self-supervised Transformer model designed for analyzing intracranial EEG recordings. It is trained using time-frequency representations of EEG data, specifically leveraging both Short-Time Fourier Transform (STFT) and superlet transform spectrograms. The model employs a masked reconstruction strategy, where random parts of the spectrogram are masked and the model learns to predict the missing portions from the surrounding context. BrainBERT is pretrained on 43.7 hours of unannotated neural recordings, providing robust and reusable neural embeddings.

- MBrain (Cai et al., 2023). MBrain is a multi-channel self-supervised learning framework designed to pre-train both EEG and SEEG signals by capturing spatial and temporal correlations among channels. It leverages Contrastive Predictive Coding (CPC) to maximize mutual information and employs tasks such as instantaneous time shift, delayed time shift, and replace-discriminative learning to enhance feature representation. Extensive experiments on large-scale real-world datasets validate its effectiveness in seizure detection.

- LaBraM (Jiang et al.). LaBraM is a unified foundation model for EEG called Large Brain Model (LaBraM). LaBraM enables cross-dataset learning by segmenting the EEG signals into EEG channel patches. Vector-quantized neural spectrum prediction is used to train a semantically rich neural tokenizer that encodes continuous raw EEG channel patches into compact neural codes

- BIOT (Yang et al., 2023). BIOT is a pre-trained EEG model that can enable cross-data learning with mismatched channels, variable lengths, and missing values by tokenizing different biosignals into unified "sentences" structure. Specifically, it tokenizes each channel separately into fixed-length segments containing local signal features and then rearrange the segments to form a long "sentence". Channel embeddings and relative position embeddings are added to each segment (viewed as "token") to preserve spatio-temporal features.

- PPi (Yuan et al., 2024b). PPi is a pretraining-based model for patient-independent seizure detection that leverages SEEG data. It employs self-supervised learning tasks, including contrastive and masked modeling, to extract rich information from the SEEG signals while preserving the unique characteristics of different brain areas. The model is pretrained on a large amount of SEEG data to handle the domain shift between different patients effectively.

- Brant (Zhang et al., 2024). Brant is a foundation model designed for intracranial neural signal analysis, utilizing EEG features. It employs techniques such as contrastive learning and masked signal modeling for training. The model is pretrained on a substantial dataset of 1.01 TB of intracranial data, enabling it to capture long-term temporal dependencies and spatial correlations across channels.

- Brant-2 (Yuan et al., 2024a). Brant-2 is a foundation model for brain signals that utilizes a diverse pre-training corpus of nearly 4 TB of SEEG and EEG data from over 15,000 subjects. It integrates time and frequency information and employs techniques such as data augmentation, mask-prediction, and future signal forecasting for training. This approach enhances its robustness to data variations and its ability to generalize across various downstream tasks.

- ScatterFormer (Zheng et al., 2023). ScatterFormer is a transformer-based model designed for patient-independent detection of epileptiform discharges using multispectral EEG feature representations. It captures fine-grained, high-frequency features through invariant scattering transform and frequency-aware attention mechanisms. The model is trained using a combination of contrastive learning and masking techniques on a comprehensive dataset of EEG records, though specific details on the pre-training dataset size are not provided in the document.

# C ARCHITECTURE AND IMPLEMENTATION DETAILS

## C.1 VQGAN SPECTRAL TOKENIZER

In this section, we provide further details about the architecture of VQGAN spectral tokenizer, and its implementation and training details.

**Architecture details** The VQGAN is based on an encoder-decoder architecture. The encoder consists of a pyramid stacking of Transformer and convolutional blocks that downsamples the input spectrograms to a low-resolution latent space, while the decoders upsamples from the quantized embeddings of the latent features to original shape and outputs the reconstructed spectrograms. We use rotary positional encoding (Heo et al., 2024), which is more flexible than absolute positional encoding methods. s The model operates at a dimension of 768, utilizing convolution and transposed convolution layers with a kernel size of 4x4 and a stride of 2x2 for downsampling and upsampling, respectively. The embedding layer transforms 16x16 non-overlapping spectrogram patches using a linear layer.

Let $x, \hat{x}$ be the original and reconstructed samples by the generator $G$, and $D$ denotes the discriminator, which is a convolutional neural network of 5 convolutional layers with kernel size $3 \times 3$ and stride $2 \times 2$ that downsamples inputs to multi-scale intermediate features during forward propagation and outputs predictions for whether the input sample is generated by $G$, then output labels indicating whether the sample is predicted or generated through global average pooling and a linear classifier head. The quantization bottleneck module is a grouped residual lookup-free quantizer with latent dimension $d$. The hyperparameters of the quantizer is listed in Table 12.

**Implementation details** We use the following loss during training of VQGAN.

1. **Reconstruction loss**. It measures the difference between original and reconstructed time-frequency features. To mitigate oversmoothing of high-frequency details, we use a combination of $L_1$ and focal frequency loss (Jiang et al., 2021) $\mathcal{L}_{\text{focal}}$

2. **Perceptual Loss**. For the original and reconstructed samples $x, \hat{x}$, intermediate activations are extracted from the multiscale discriminator of M intermediate layers as $a_i, \hat{a}_i, i = 1, \ldots, M$. The perceptual loss is formulated as

$$\mathcal{L}_{\text{perceptual}} = \mathbb{E}_{x \sim p(x)} \sum_{i=1}^{M} \|a_i - \hat{a}_i\|. \tag{11}$$

3. **Generator Loss**. It measures the dissimilarity between reconstructed and original samples:

$$\mathcal{L}_{\text{generator}} = -\mathbb{E}_{\hat{x} \sim p(\hat{x})}[\log D(\hat{x})] \tag{12}$$

4. **Discriminator Loss**. It is used to train the discriminator for distinguishing reconstructed and original samples.

$$\mathcal{L}_{\text{discriminator}} = -\mathbb{E}_{x \sim p_{\text{data}}(x)}[\log D(x)] - \mathbb{E}_{\hat{x} \sim p(\hat{x})}[\log(1 - D(\hat{x})] \tag{13}$$

5. **Commitment Loss**. Let $z, \hat{z}$ be the latent representations before and after quantization, the commitment loss $\mathcal{L}_{\text{commitment}}$ are defined as

$$\mathcal{L}_{\text{commitment}} = \mathbb{E}_{x \sim p(x)} \sum_{i=1}^{d} \left\| z - \text{sg} \left[ \hat{z}^{(i)} \right] \right\|_2^2, \tag{14}$$

where sg[·] is the stop-gradient operator, and the straightthrough estimator is used for the backpropagation through the quantization module. Note that $\mathcal{L}_{\text{commit}}$ is the sum of quantization errors from every $i = 1, 2, \ldots, d$. It aims to make $\hat{z^{(i)}}$ sequentially decrease the quantization error of $z$ as $i$ increases. Thus, it approximates the feature map in a coarse-to-fine manner and keeps the training stable. To encourage codebook utilization, we add an entropy regularization term

$$\mathcal{L}_{\text{entropy}} = \mathbb{E}_{x \sim p(x)} \sum_{i=1}^{d} \mathbb{E} H(\hat{z}^{(i)}) - H(\mathbb{E} \hat{z}^{(i)}), \tag{15}$$

where $H$ is the entropy function.

| Architectural Hyperaramaters | Value |
|---|---|
| group | 4 |
| latent dimension | 512 |
| codebook size | 16384 |
| depth | 8 |

Table 12: **Architectural hyperparameters of the quantization bottleneck**.

| Architectural Hyperaramaters | Value |
|---|---|
| Embedding Dimension | 768 |
| Number of Heads | 16 |
| Number of Spectral Transformers | 12 |
| Number of Multi-View Transformers | 4 |
| Number of Decoder Transformers | 4 |
| Expanding Factor in SwiGLU | 4 |

Table 13: **Architectural hyperparameters of *Beatrix***.

**Training Details** The network is implemented in PyTorch 2.1.0. The code is available in supplementary materials. We use AdamW optimizer with $\beta_1 = 0.9, \beta_2 = 0.99$ and weight decay of 1e-4 during training. The generator loss is formulated as

$$\mathcal{L} = \mathcal{L}_{\text{generator}} + \lambda_1 \mathcal{L}_{\text{commit}} + \lambda_2 \mathcal{L}_{\text{entropy}} + \lambda_3 \mathcal{L}_{\text{perceptual}} + \lambda_4 \mathcal{L}_{L_1} + \lambda_5 \mathcal{L}_{\text{focal}}, \quad (16)$$

where $\lambda_1 = 0.1, \lambda_2 = 0.01, \lambda_3 = 0.05, \lambda_4 = 2.0, \lambda_5 = 2.0$. A cosine annealing scheduler with 1k warm-up steps is used to adjust learning rate. The network is trained for 10k steps.

## C.2 MAIN ARCHITECTURE

**Architecture details**

Beatrix is architected with a spectral Transformer composed of M layers of Transformer encoders and a multiview Transformer comprising N layers. The generative reconstruction of spectrograms is handled by a decoder consisting of P layers of Transformer blocks. The spectral Transformer operates by applying self-attention to tokens within each channel in isolation, whereas the multiview Transformer facilitates attention across tokens that share the same time-frequency location across various channels. This design efficiently reduces computational expenses while capturing interchannel correlations. It is important to note that the decoder operates independently for each channel and is not engaged during the fine-tuning phase. In addition, we use rotary positional encoding (Heo et al., 2024), which is more flexible than absolute positional encoding methods

Table 13 shows the architectural hyperparameters of *Beatrix*.

**Implementation details** The model is implemented using PyTorch 2.1. In particular, we use FlashAttention and SwiGlu implemented using Xformers to accelerate running speed.

**Further discussion about semi-causal generative pre-training** Mainstream generative pre-training currently relies on the sequence modeling paradigm, wherein data from diverse modalities undergo tokenization to form sequential data. Within this framework, various sequence modeling methods have been developed.

Causal sequence modeling stands out for its robust capabilities in zero-shot generalization and in-context learning, attributed to its high sample efficiency. This approach leverages the participation of all tokens in prediction, thereby providing comprehensive supervision information. It has demonstrated state-of-the-art (SOTA) performance across domains such as general-purpose natural language generation, reasoning, decision-making, and has found applications in vision and audio modeling. On the other hand, non-causal sequence modeling, based on an encoder-decoder architecture, fosters transferability across tasks and modalities while achieving enhanced fine-tuning efficiency. Widely employed in language understanding, sentiment analysis in natural language processing, as well as image and video classification, segmentation, and reconstruction in vision tasks, this method has been the cornerstone of previous EEG foundation models. Semi-causal sequence modeling rep-

resents a hybrid approach that combines elements of both causal and non-causal methods. Here, the model is tasked with predicting each token outside the prefix given all preceding tokens.

Prior work on EEG foundation models, exemplified by references such as (Jiang et al.; Zhang et al., 2024; Yuan et al., 2024b), predominantly adopts non-causal sequence modeling. However, it is worth noting that the bidirectional attention mechanism utilized in non-causal modeling suffers from a rank degeneration issue, where representations with degenerated ranks become indistinguishable due to the absence of informative components. Conversely, unidirectional attention, while less effective than its bidirectional counterpart in capturing global context information, offers a solution to the rank degeneration problem. Thus, we advocate for the adoption of semi-causal sequence modeling, which encompasses both autoregressive and non-autoregressive prediction of masked and subsequent tokens.

### C.3 TEMPORAL TOKENIZER

The temporal tokenizer used in CIFT is a one-dimensional convolutional neural network of 5 layers, each convolutional layer has a kernel size of 3 and a stride size of 2. The temporal tokenizer, similar to the spectral tokenizer, process different EEG channels independently. Through a linear projection layer, all the output feature embeddings are simply concatenated sequentially, which will be used to calculate temporal prompts through cross attention mechanism.

## D TRAINING DETAILS

To make the model adaptive to variable input signals, we select the context length of Beatrix uniformly from $L = \{2s, 4s, 6s, 8s, 10s, 16s\}$. Within non-causal spans we mask 75% of the total tokens. For a sequence we sample $M \in [0, L-1]$ uniformly, $L$ is the total length, and $M$ is the length of the non-causal span. To make the training process simple, we designate the first $M$ tokens for non-causal modeling, and the rest are used for causal modeling. The channel number of EEG samples is uniformly chosen between 1 and $C$, $C$ is the maximal number of utilizable channels for each record. The model is pre-trained for a total of 100,000 steps on 4 GPUs (NVIDIA Tesla A100 40G). The optimizer is AdamW with $\beta_1 = 0.99, \beta_2 = 0.999$ and a weight decay rate of 1e-4 and initial learning rate of 1e-6. A cosine annealing scheduler is used with a total of 5,000 warm-up steps and maximum learning rate of 1e-4. At fine-tuning stage, we use 100 warmup steps and a maximum learning rate of 1e-5 on one GPU.

## E DATASETS

In this section, we present a comprehensive list of datasets utilized in both pre-training and fine-tuning, along with necessary details. We also present the data collection and cleansing procedure which pays a special attention on privacy and fairness in training of foundation model.

For the public datasets, we provide downloading links. For the private dataset used in this work, we provide the preprocessed, anonymized data in an anonymous link for reproducibility and further research. The full, raw recordings used for contructing the private dataset are available upon reasonable request.

### E.1 DATA COLLECTION AND CLEANSING

The diversity and quality of the data utilized significantly influence the training process and subsequent performance on downstream tasks. Therefore, it is crucial to curate a comprehensive collection of data to train the foundation model on a wide array of datasets.

During the pre-training stage, we gather a substantial amount of openly available neuroelectrophysiological signals, with a particular focus on EEG data. While EEG datasets are increasingly available online, large-scale datasets are still relatively scarce due to the costs associated with data recording and privacy concerns. To ensure a diverse and representative corpus that encompasses various subjects, diseases, and tasks, we carefully curate the collected datasets. Unlike previous works that predominantly rely on clinical resting state EEG records of patients and normal controls,

we extensively gather data from a range of diseases and normal states, including resting state and task/event-evoked brain activities. Some datasets are allowed to be downloaded freely, others need registration.

Uncensored pre-training corpora often contain numerous corrupted data records due to discrepancies in data recording and preprocessing conditions. To address this, we strictly exclude unusable or corrupted data and harness about 32,900 hours of EEG data for pre-training. We checked the data using EDFBrowser and MATLAB EEGLAB and excluded the unusable sessions manually before preprocessing. Sinc interpolation is used for resampling and Notch filter with a quality factor of 50 and an order of 2 is used at 50 or 60 Hz.

We ensure all data are anonymized and de-identified and store them in FIF data format using Python MNE to avoid loss of floating-point precision.

Given the lack of consensus on the sampling and segmentation of EEG data, during training, we load each data record into memory as a whole and randomly segment it into epochs of varying lengths. Furthermore, we resample the data to 256Hz and apply a 50 and/or 60 Notch filter to filter out utility frequency noise.

To alleviate distributional bias in the pre-trained model, we rebalance the data distribution of the train dataloader based on the total number of subjects in each dataset.

### E.2 PRE-TRAINING EEG CORPUS

We provide a comprehensive list of publicly available datasets used to construct the EEG corpus for pre-training our models as follows.

- **TUH EEG Corpus**. This dataset contains 26,846 clinical EEG recordings collected at Temple University Hospital (TUH) from 2002 to 2017, covering health and disease conditions.

  Link: https://isip.piconepress.com/projects/tuh_eeg/html/downloads.shtml.

- **TDBRAIN**. This dataset contains resting-state, raw EEG-data complemented with relevant clinical and demographic data of a heterogenous collection of 1274 psychiatric patients collected between 2001 to 2021.

  Link: https://brainclinics.com/resources/

- **Neuroforecasting**. This dataset contains EEG data recorded in studying neural activity during value-based decision-making. It involves resting-state and visually-evoked EEG activities associated with individual choice and market outcomes.

  Link: https://openneuro.org/datasets/ds004284/versions/1.0.0

- **HD-EEGTask**. This dataset contains task-evoked EEG during visual object naming and spelling tasks.

  Link: https://openneuro.org/datasets/ds003420/versions/1.0.2

- **DepressionRest**.This dataset contains resting-state EEG data with 122 college-age participants. Task included in DMDX programming language, with instructions for eyes open and eyes closed triggers.

  Link: https://openneuro.org/datasets/ds003478/versions/1.1.0

- **TBI**. This dataset contains EEG records of traumatic brain injuries (TBIs), involving three stimulus auditory oddball data in control, sub-acute mild TBI, and chronic TBI. Rest-state data is also included.

  Link: https://openneuro.org/datasets/ds003522/versions/1.1.0

- **Improvision and Music Structures**. This dataset contains EEG data recorded during a study of musicians' brain responses to chords, involving resting-state and audio-evoked activities in different tasks.

  Link: https://openneuro.org/datasets/ds003570/versions/1.0.0

- **Reward Biases**. This dataset contains EEG data recorded during sleep EEG for 1-2 hours in participants who played at 2 different games during wakefulness.

Link: https://openneuro.org/datasets/ds003574/versions/1.0.2

- **Verbal Working Memory**. This dataset contains EEG records in a modified Sternberg working memory paradigm with two types of task: with mental manipulations (alphabetization) and simple retention (TASK) and 3 levels of load: 5, 6, or 7 letter to memorize (LOAD).

  Link: https://openneuro.org/datasets/ds003655/versions/1.0.2

- **Social Memory Cuing**. This dataset contains EEG records from subjects who participate in a memory task presented in virtual reality.

  Link: https://openneuro.org/datasets/ds003702/versions/1.0.1

- **CSA**. This dataset contains EEG records in a neurobehavioral study on women with childhood sexual abuse and problem drinking at Washington University.

  Link: https://grantome.com/grant/NIH/R01-AA025646-04

- **Trance Channeling**. This dataset contains 13 participants that went through a thorough screening and did 2 sessions (different days) each. Experiment design corresponded in alternating (5 minutes) blocs of trance channeling and resting state (3 periods per session for each condition).

  Link: https://openneuro.org/datasets/ds004040/versions/1.0.0

- **SRM**. This dataset contains resting-state EEG extracted from the experimental paradigm used in the Stimulus-Selective Response Modulation (SRM) project at University of Oslo.

  Link: https://openneuro.org/datasets/ds003775/versions/1.2.1

- **Resting and Cognitive States**. This dataset contains resting(eyes closed, eyes open) and cognitive(subtraction, music, memory) state EEG recordings with 60 participants during three experimental sessions together with sleep, emotion, mental health, and mind-wandering related measures.

  Link: https://openneuro.org/datasets/ds004148/versions/1.0.1

- **Sound Source Elevation**.The dataset consists of data from two experiments in which subjects were presented bursts of noise from loudspeakers at different elevations. Subjects who participated in either experiment were initially tested in their ability to localize elevated sound sources. Both experiments were conducted in a hemi-anechoic chamber.

  Link: https://openneuro.org/datasets/ds004256/versions/1.0.5

- **HBN**. This dataset 2952 children's eyes-open and eyes-closed EEG. Eyes-open lasted for 20 seconds, and eyes closed for 40 seconds.

  Link: https://openneuro.org/datasets/ds004186/versions/2.0.0

- **Reversal Learning**. This dataset contains EEG records during two reversal learning tasks with different reinforcer (monetary reward versus primary threat reinforcer). Positive feedback in the reward task indicated monetary reward and negative feedback monetary non-reward.

  Link: https://openneuro.org/datasets/ds004295/versions/1.0.0

- **Large Spanish EEG**. This dataset contains EEG responses to silent and perceive speech on 30 spanish sentences.

  Link: https://openneuro.org/datasets/ds004279/versions/1.1.1

- **Executive Functioning Tasks**. This dataset contains task-evoked EEG data in executive functioning battery consisting of three separate tasks: 1) N-Back (NB); 2) Sustained Attention to Response Task (SART); 3) Local Global (LG).

  Link: https://openneuro.org/datasets/ds004350/versions/1.1.1

- **PEERS**. This dataset contains EEG records in a study on the behavioral and electrophysiological (EEG) correlates of memory encoding and retrieval in highly practiced individuals. Across five experiments, more than 300 subjects contributed more than 7,000 90 minute memory testing sessions.

  Link: https://openneuro.org/datasets/ds004395/versions/2.0.0

- **Continuous Naturalistic Speech**. This dataset contains EEG responses of healthy, neurotypical adults who listened to naturalistic speech. The subjects listened to segments from

an audio book version of "The Old Man and the Sea" and their brain activity was recorded using a 128-channel ActiveTwo EEG system (BioSemi).

Link: https://openneuro.org/datasets/ds004408/versions/1.0.8

- **Vicarious Touch**. This dataset contains EEG records with and without vicarious touch experiences to test whether seen touch evokes overlapping neural representations with the first-hand experience of touch. Participants felt touch to the fingers (tactile trials) or watched carefully matched videos of touch to another person's fingers (visual trials).

  Link: https://openneuro.org/datasets/ds004563/versions/1.0.1

- **Normal Infants** This dataset contains resting EEG for a sample of 103 normal infants (41 female and 62 male) in the first year of life.

  Link: https://openneuro.org/datasets/ds004577/versions/1.0.1

- **Neuma**. This dataset contains multi-modal brain data from 42 individuals who participated in an advertising brochure-browsing scenario is introduced here. In more detail, participants were exposed to a series of supermarket brochures (containing various products) and instructed to select the products they intended to buy. The data collected for each individual executing this protocol included: 1) encephalographic (EEG) recordings, 2) eye tracking (ET) recordings, 3) questionnaire responses (demographic, profiling and product related questions), and 4) computer mouse data.

  Link: https://openneuro.org/datasets/ds004588/versions/1.2.0

- **Infant Microstate Reliability**. This dataset contains EEG records from infants watching video.

  Link: https://openneuro.org/datasets/ds004635/versions/3.0.0

- **ERP**. This dataset contains EEG data recorded in a multi-site study of event-related brain potential (ERPs) and their task-specific relationships.

  Link: https://openneuro.org/datasets/ds004602/versions/1.0.1

- **Python Reading Task**. This dataset contains EEG data records during Python code reading.

  Link: https://openneuro.org/datasets/ds004771/versions/1.0.0

- **TNO**. This dataset contains task-evoked P300 responses.

  Link: https://openneuro.org/datasets/ds004660/versions/1.0.2

- **Loneliness**. This dataset contains EEG data recorded in a study on distinguishing how lonely individuals respond to negative social stimuli in a roving oddball paradigm.

  Link: https://openneuro.org/datasets/ds004802/versions/1.0.0

- **Music Therapy**. This dataset contains EEG data recorded from adult burn patients in the intensive care unit during music therapy.

  Link: https://openneuro.org/datasets/ds004840/versions/1.0.1

### E.3 MULTI-CENTER INTRACRANIAL EEG DATASET

We construct a benchmark from intracranial EEG records sourced from National Institute of Health (NIH), University of Maryland Medical Center (UMMC), Johns Hopkins Hospital (JHH), and University of Miami Florida Hospital (UMFH) (Li et al., 2023). It poses a significant challenge for generalizable, subject-independent seizure forecasting up to 5 minutes in advance. The individual variability in epileptogenic lesions, discerned through MRI scans and post-surgical treatment outcomes, contributes to this complexity. Additionally, the diversity in recording conditions and electrode montages across these hospitals makes the dataset highly heterogeneity. Fine-tuning and model selection use data from NIH, UMMC, and JHH, while reserving the UMFH data for evaluation.

### E.4 SELF-COLLECTED DATASET

**Dataset description and preprocessing** Our dataset stands out for its comprehensive collection of resting-state EEG data from patients exhibiting a range of epilepsy subtypes, including clonic, absence, and tonic seizures. With 16 participants diagnosed with absence seizures, 5 with clonic

seizures, and 6 with atonic seizures, our dataset offers a rich tapestry of well-annotated EEG recordings. This dataset is particularly valuable because, to our knowledge, no other benchmark exists that provides such a diverse and meticulously annotated set of EEG data for these relatively uncommon but clinically significant epilepsy subtypes for EEG domain generalization. While the famous TUH EEG Corpus does encompass various seizure patterns, the uniqueness of our dataset is amplified by the distinct geographical origins and genetic backgrounds of our participants, markedly different from those in the TUH EEG Corpus.

Our experimental protocol involves continuous monitoring of patients across both awake and sleep stages, enabling a more exhaustive observation of ictal and non-ictal events. This approach captures the necessary intra-subject variability, which is crucial for understanding the dynamics of epileptic dynamics.

Our dataset includes annotations of epileptiform discharges for both ictal and interictal stages, spanning conscious and sleep states. By extracting samples with a 2-second window and a sampling rate of 256 Hz, we have constructed a challenging benchmark for out-of-distribution (OoD) seizure detection across different disease types.

In the main body of the paper, we concentrate on domain generalization between clonic and abtonic seizures, which represent two mechanistically distinct forms of epilepsy. Additionally, we present supplementary experiments in the appendix, exploring other potential scenarios to further demonstrate the versatility and robustness of our dataset.

**Ethical statement and data availability** The collection and use of this data have been rigorously reviewed and approved by the hospital's ethics committee, with all participants providing their informed consent, ensuring the ethical standards are maintained throughout our research. The preprocessed samples are available for further development of OoD EEG algorithms. The raw data is available upon request and we are working to formally publish it in the future.

## F  MORE RELATED WORK

**OoD Generalization and Fine-Tuning** Various fine-tuning methods have been developed to efficiently adapt pre-trained models to new tasks with minimal parameter adjustments. However, standard fine-tuning can compromise OoD generalizability (Li et al., 2024; Lee et al., 2023; Kumar et al., 2022; Wortsman et al., 2022; Andreassen et al., 2021). Salman et al. demonstrate that fine-tuning can degrade pre-trained features, adversely affecting OoD performance (Salman et al., 2020). New techniques have emerged to counteract these effects (Lee et al., 2022; Wortsman et al., 2022; Kumar et al., 2022). Parameter-Efficient Fine-Tuning (PEFT) strategies, such as those illustrated by Lee et al. and Kim et al., effectively mitigate distribution shift issues (Lee et al., 2023; Kim et al.). Empirical evidence suggests that parameter-efficient fine-tuning, often employing adaptors (Sung et al., 2022; He et al., 2021; Houlsby et al., 2019), enhances OoD generalization, especially when downstream data is limited (Chen et al., 2024; Goyal et al., 2023; Zheng et al., 2022). Prompt tuning, a variant of PEFT, introduces flexible trainable prompts for multi-modal extensions (Li et al., 2024; Samadh et al., 2023; Shu et al., 2022).

