# OpenReview forum: "Beatrix: Out-of-Distribution Generalization of Large EEG Model via Invariant Contrastive Fine-Tuning"
_ICLR.cc/2025/Conference — Submitted to ICLR 2025_

### Official Review · Reviewer_KRhy · 2024-10-21

**Soundness:** 2
**Presentation:** 1
**Contribution:** 1
**Rating:** 1
**Confidence:** 4

**Summary:**

This paper presents Beatrix, an electroencephalogram (EEG) foundation model to enhance out-of-distribution (OOD) generalization. It employs the semi-causal generative approach for pre-training and the Contrastive Invariant Fine-Tuning (CIFT) method to facilitate domain-invariant learning without explicit environment labels. Experiments in the main paper and appendix demonstrate the performance of Beatrix in several tasks related to EEG analysis.

**Strengths:**

Extensive experiments conducted on multiple datasets and tasks, such as seizure detection, motor imagery, and sleep staging, offer comprehensive evidence of the model's performance.

**Weaknesses:**

- **Lack of Novelty**: All the components, including analytic wavelet transform, semi-causal generative pre-training, and VQGAN, have been previously proposed in the literature. The paper describes a lot for these parts, but it does not give correct citations. It seems to be packaged as its own work. For the remaining component, CIFT, there are factual errors and unclear representations in the description of the method.  Moreover, it is counter-intuitive to apply bottleneck adapters and low-rank adapters in the same model, which is not explained in the paper. Hence, the novelty of the paper is weak.
- **Factual Errors**:
1.  L262, 'CIFT incorporates three distinct low-rank adapters' and the following describes 'BottleNeck adaptor', 'Lora adaptor', 'Layer Normalization', where the 'Layer Normalization' is not a low-rank adapter but a common component in Transformer models.
2.  L212 admits that the model is based on Transformer, while the architecture in Figure 3 lacks the Feed-Forward component.
3.  In Figure 4, the right part of the figure is not clear. In particular, there are two arrows pointing to nothing and the computation of cross-entropy loss is not clear.
- **Experiments**:
1. The main experiments compared with the state-of-the-art methods and ablation study are arranged in the appendix. The main paper should include the most important experiments and results.
2. The paper could have provided comprehensive ablation studies on CIFT, e.g. ablation on w/ or w/o CLIP loss.
- **Representation Problem**:
1.  All the Figures are not vector diagrams, which are not formal.
2.  L141-143, the meaning of e is not clear.
3.  L237, the meaning of semi-causal is not clear. What is the difference between semi-causal and causal?
4.  A redundant left bracket in L250.
5.  Caption of Figure 3 dismatch the content of the figure.
6.  The words 'adaptor' and 'adapter' are used interchangeably, which is not formal.
7.  L377, 'beta' should be \beta.

**Questions:**

Discussed in Weaknesses.

---

> ### Author Response · Authors · 2024-11-26
>
> We thank the reviewer for the helpful feedback.
>
> “Lack of Novelty”
>
> Our work introduces significant novelties that set it apart from prior approaches. We combine spectral tokenization with generative pre-training and invariant contrastive fine-tuning (CIFT) to improve domain-invariant embeddings and OoD performance. We conducted extensive experiments across multiple datasets to demonstrate robust OoD performance. We propose novel approaches that are likely to influence future research in EEG data analysis and biomedical signal processing.We curated an extensive list of open EEG data corpora (arguably the largest one to our knowledge) for pretraining, which enhances the reproducibility of this work and will facilitate future research on EEG foundation models.
>
> We would like to highlight that all previous works were cited in the initial version across different sections of the main text. Bottleneck and low-rank adapters are compatible: the bottleneck adapter operates following the main network module, while the low-rank adapter is integrated within it. We refined these in the revised version.
>
> “Factual Errors”
>
> 1. The term "low-rankness" refers to parameter-efficient adapters, as clarified in the main text. This phrasing is refined in revised version.
>
> 2. We clarify that we use SwiGLU throughout the work as our Feed-Forward component, which is explicitly denoted in Figure 3, see main text discussion in Sec. 3.2
>
> 3. We acknowledge the issue with the unclear arrows and rectified it by ensuring a clear and closed connection between the two lines in the revised version.
>
> Experiments
>
> We structured the paper to focus the main text on key contributions and results while placing detailed analyses in the appendix due to space constraints. We would like to highlight that we already ablated it in Table 1, with ERM method (which means no additional OoD auxiliary losses are used except for the cross-entropy loss), i.e., w/ and w/o CIFT loss. We also compared CIFT with other contrastive OoD generalization algorithms.
>
>
> Representation Problem
>
> 1. This will be revised in the final version.
>
> 2. We clearly pointed out in Sec. 2.1 that e represents an environment or domain.
>
> 3. Semi-causal refers to a hybrid pretraining framework that combines causal (unidirectional, using past tokens to predict future ones) and non-causal (bidirectional, using both past and future tokens for reconstruction) spans. Unlike purely causal models, semi-causal modeling enables both extrapolation and interpolation across temporal and spectral domains. Previous works only use non-causal approach. Our approach balances temporal consistency with broader contextual understanding, enhancing the model’s ability to generalize to complex EEG dynamics.
>
> Other details have been refined in the revised version.

---

> > ### Comment · Reviewer_KRhy · 2024-11-27
> >
> > Regarding novelty, the claimed techniques involve pre-training a foundation model and fine-tuning it using contrastive loss techniques. However, most techniques, including PEFT, analytic wavelet spectral analysis, etc., are not proposed in this work, occupying too much space and leading to a bad representation of the inherent contribution.  It would be beneficial to provide a thorough explanation and showcase the uniqueness of the 'Environment-Aware Reweighting' section.
> >
> >
> >
> > Furthermore, in terms of motivation, Lines 56-57 assert that only a small number of models utilize time-frequency. Lines 487-489 acknowledge the existence of models pre-trained on time-frequency, indicating a weak motivation for pre-training a new foundation model on time-frequency.
> >
> >
> >
> > Regarding the experiment, I continue to recommend viewing the comparison with previous foundation models as the primary focus. The table contains a significant amount of baseline data without citations, such as HEX and EnD, which can make it challenging for readers to understand the background.
> >
> >
> >
> > The last row of Table 3 should be SD+TD, if I understand correctly.
> >
> >
> >
> > Regarding representation, previous reviews have identified numerous errors that have not been rectified in the revised version. For instance, the figures are still in PNG format, while the beta version is in L384.
> >
> >
> >
> > I recommend a thorough reorganization of this word and increased effort from the authors to address the aforementioned issues.

---

> > > ### Author Response · Authors · 2024-11-28
> > >
> > > We thank the reviewer for the detailed reply.
> > >
> > > (1) While techniques like PEFT and wavelet analysis are part of the overall methodology, they are not the focus of the paper. Instead, they serve as tools within a novel framework that is uniquely tailored to EEG data. We believe that the existing implementation details do not overshadow our core contributions but rather serve to enhance and validate them.
> > >
> > > The ‘Environment-Aware Reweighting’ section is a distinct contribution of the paper, and we have already clarified its role in improving OoD generalization. This method allows the model to weigh environmental factors without explicit supervision, an approach not commonly used in EEG research. We will further emphasize its uniqueness in the revised manuscript, with additional explanations and examples to showcase its novelty.
> > >
> > > (2) The motivation for using time-frequency representations is grounded in the unique nature of EEG signals, which exhibit both transient (time-domain) and oscillatory (frequency-domain) behaviors. While it is true that some models use time-frequency representations, the key innovation here is in the integration of both domains into a unified framework, which has shown to improve generalization across multiple EEG tasks. This combined representation is not common in existing literature, and its effectiveness is demonstrated in our ablation studies and experiments.
> > >
> > > (3) We clarify that OoD baselines were cited in the initial version in the Experimental Setup section.
> > >
> > > (4) We agree with the reviewer that further refinement would enhance the quality of the paper, and we have made these improvements in the revised version.

---

### Official Review · Reviewer_hmwf · 2024-11-01

**Soundness:** 1
**Presentation:** 2
**Contribution:** 3
**Rating:** 3
**Confidence:** 3

**Summary:**

The submitted manuscript studies the problem of training and fine-tuning of electroencephalography (EEG) foundation models for out-of-distribution generalization (here, across subjects and sub-tasks).
Building upon prior works on EEG foundation models and invariant risk minimization, the authors propose a three-stage modeling strategy.

They split foundation model pre-training into two steps.
Using a corpus of publicly available diverse EEG datasets, they first train a tokenizer model that transforms short segments of single-channel spectrogram data into tokens.
They use a combination of several unsupervised loss terms (e.g., reconstruction, quantization, ...) to compose their training objective.
Thereafter, they use the same EEG corpus to train a combination of spectral and multi-view transformer networks.
The former is constrained to mix tokens within each spatial channel, and the latter is constrained to mix tokens along the channel-dimension independently for each time-frequency bin.
As loss they employ a reconstruction loss for masked tokens.
The employed semi-causal masking strategy combines BERT-style random masking and reconstruction with GPT-style autoregressive reconstruction.
After pre-training is completed most of the model parameters are frozen.

For model fine-tuning they propose contrastive invariant fine-tuning (CIFT) to adapt the model to specific downstream tasks, primarily, seizure detection and forecasting.
CIFT combines several ideas to perform parameter-efficient fine-tuning (PEFT) with a domain generalization learning objective.
First, a combination of prior PEFT methods (LoRA and bottleneck adaptors, trainable layernorm parameters) are utilized to tune the parameters of the spectral and multi-view transformer network.
Additionally, a  temporal embedding layer is introduced along with trainable prompts.
A cross-attention layer is used to summarize the spectral and temporal information of each segment in the trainable prompts.
After passing the resulting tokens through the network, the proposed loss terms are computed.
Apart from the cross-entropy loss, the CIFT loss encompasses a CLIP loss between the spectral and temporal prompts that is tuned to focus on environment-invariant latent representations.
The authors evaluate their proposed approach in several experiments and ablation studies with EEG data.
Their evaluation focuses on seizure detection (n=16 subjects) and seizure forecasting (n=35 subjects) problems. Compared to many baseline methods, the proposed approach obtains the highest performance scores.

**Strengths:**

### Originality

The considered problem setting that tests for out-of-distribution generalization is a challenging yet highly relevant setting.
The proposed framework employs a domain generalization approach based on invariance minimization that has been under-explored in neuroimaging data including EEG. In combination with data-driven estimation of environment labels, the approach seems appealing for many practical use cases.

### Quality

Key to the success of EEG foundation models is to combine temporal/spectral/spatial integration in the presence of diverse hardware and electrode configurations.
The presented empirical results indicate that this contribution proposes a suitable solution for the problem.
Additionally, the presented ablation studies touches on some relevant directions.

### Clarity

The authors provide sufficient motivation for most of the design choices so that their combination of ingredients generates a plausible recipe to address the problem setting.

### Significance

Combining a foundation model with an invariant risk minimization fine-tuning objective to address the EEG domain generalization problem is a novel and interesting approach.

**Weaknesses:**

### Quality of the Experiments

- The presented empirical results (Table 1 to 11 as well as Figure 5) summarize the average performance across test sets, yet fail to indicate the variability of the results across random seeds and subjects. The lack of reporting variability is problematic because the considered datasets are of relatively small size. For example, the results presented in the main text (Table 1 to 2) consider datasets with a total of 16 to 35 subjects. Since subsets (whose sizes are not reported) are used for testing, the actual number of subjects in the test set are likely considerably smaller.
- I appreciate that the authors tested the proposed CIFT approach also with different model architectures (Table 2). However, after carefully studying the ablation experiments I am still wondering what factors drive the success of the proposed approach.
Unlike stated in lines 463 to 466, the particular choice of fine-tuning adapter parameters seems to have a large impact on performance (Table 6).
In the same paragraph (lines 463 to 466) the authors mention that full-parameter and linear probing yielded similar OoD generalization as the proposed combination of adapter methods without substantiating the claims in terms of simulation results in tabular of graphical form.
- There seems to be a discrepancy between the performance of the proposed method (Beatrix + CIFT (r=16, K=8)) as summarized in Table 1 and the results summarized in Figure 5 (K=8, r={4,8,12,16}) at r=16. The results in Figure 5 are substantially lower (e.g., for the seizure forecasting dataset below 90% accuracy (blue line @ rank=16) compared to 91.8% reported in Table 1).
- The additional experiments on auditory brain decoding and motor imagery contain too little information to properly assess the presented results.
For example, the motor imagery experiments only comprise of Table 9 without any indication of the considered dataset.


### Clarity of Contrastive Invariant Fine-tuning (CIFT)

- It is unclear why you introduce a CLS token. Even though it is introduced it is not used in the considered loss functions.
- The function $\rho$ and its learnable parameters $\eta$ are not defined.
- The dimensionalities of the symbols in equations (5) to (10) are not defined.
- You use the undefined function $\rho$ to estimate the environment $\hat{e}$ and class $\hat{y}$ labels. However, in the considered datasets the number of classes is potentially different from your choice of virtual environment dimensionality $K$.

**Questions:**

### Introduction

- The first paragraph uses the terms neuroelectrophysiology and EEG (which is not defined but I assume you refer to electroencephalography) interchangeably. This is imprecise since both terms refer to different (but somewhat related) concepts.
- Please use either "invariant contrastive fine-tuning" (title) or "contrastive invariant fine-tuning" (main text).

### Preliminaries

- $X^e$ and $Y^e$ in (1) are not defined.
- Readers who are not familiar with invariant risk minimization might not be familiar with the training objective, defined in (1). Please refer to relevant prior works.

### Methods

- lines 190 to 195: please additionally specify the dimensionality of the generated wavelet features as well as the patch size (as visually indicated in Figure 2.)
- Which probabilistic model did you assume in equation (4). If you used an isotropic Gaussian model, the NLL loss reduces to the classical MSE loss. Please clarify.
- $\mathrm{Aggregate}(\hat{V})$ in (7) should be $\mathrm{DifferentialbeHeaviside}(\hat{V})$
- dimensionalities of $\hat{e}$ and $l_{\mathrm{CIFT}}$ in (10) are unclear


### Experiments
- I would like the authors to clarify the differences between results in Table 1 and Figure 5.
- To address my concerns about variability, the authors should run additional experiments with more random seeds and report the obtained standard deviations (across seeds, but ideally also across subjects). Beyond that, the authors should conduct significance testing at the level of subjects. That is they should compute the performance of the model for each subject in the test set and then compare the results across baseline models with appropriate significance tests (or at least report test statistics like t-values).
- Since the authors report that the temporal domain tokenizer is an important ingredient in their method, they should also substantiate their claim with an appropriate ablation experiment.
- Please explicitly report the number of subjects and classes in the dataset as well as how you generated the data splits and their sizes (# subject and samples).
- Table 1: category SF is undefined (and overlaps with the scenario seizure forecasting (SF)). Column AUPRC/SD has formatting issues.

### Wording, Grammar and Organization

- line 159: SEEG is not defined.
- line 222: "adds" -> "add"
- line 230: "arrange" -> "range"
- line 331: "is" -> "are"
- line 377: $beta$ -> $\beta$
- line 430: define PEFT
- line 431: define ERM
- line 558: there is a broken reference so the appendix.
- line 664 to 666: the bibliography entry is incomplete.

---

> ### Author Response · Authors · 2024-11-26
>
> We thank the reviewer for the helpful feedback.
>
> Quality of the Experiments
>
> 1. The experiments are conducted following standard practices in EEG foundation model research, where we report the mean performance across multiple test sets. Some baseline pretrained checkpoints were unavailable due to intellectual property and privacy concerns, which restricted our ability to conduct statistical tests.
>
> 2. We clarify that full-parameter and linear probing results were presented in Table 1 as ERM-Full and ERM-LP, respectively. Moreover, choice of rank has significant influence on the performance. Given limited computational budget, larger rank leads to better performance. Ablation of different adapters in CIFT was presented in the appendix.
>
> 3. Regarding the visual discrepancy between the performance reported in Table 1 and Figure 5, it is not due to differences in the underlying data, but rather an artifact in the visual representation in Figure 5, which arose from the way the graph was rendered in Python using Matplotlib.
>
> 4. We have refined the details in the revised version.
>
> Clarity of CIFT
>
> We have addressed and clarified the ambiguities in the CIFT section in the revised version, providing specific clarifications in response to the reviewer’s points as follows.
>
> 1. We clarify that the [CLS] token is not used in CIFT or any other OoD generalization algorithms presented in this work. This has been corrected in the revised version.
>
> 2. \(\rho\) is defined as an auxiliary function for estimating virtual environment labels for each sample. It is not used during the inference stage of CIFT. The additional parameters \(\eta\), introduced by \(\rho\), are implemented via a two-layer MLP, as already noted in the main text.
>
> 3. We highlight that the dimensions for the contrastive fine-tuning loss were provided in the corresponding subsection.
>
> 4. We clarify that \(\rho\) is solely used to estimate virtual environments \(\hat{e}\), not class labels \(\hat{y}\). Furthermore, the number of virtual environments \(K\) is unrelated to the number of classes, and this distinction is now clearly articulated.
>
> Questions
>
> Introduction
>
> We have polished the introduction section to avoid ambiguities.
>
> Preliminaries
>
> 1. It is defined as a sample and its label belonging to environment \( e \) within a potential set of environments.
>
> 2. Relevant works were cited in the OoD baselines section, as well as additional related work section in the appendix in the initial version. Additional citations have been included in the revised version for completeness.
>
> Methods
>
> 1. We clarify that for a 4-second EEG sample at a sampling rate of 256 Hz, raw wavelet features are downsampled to \( 64 \times 256 \), with a patch size of \( 8 \times 8 \) adopted.
>
> 2. The issue has been corrected in the revised version.
>
> 3. We highlight that \(\hat{e}\) represents the predicted virtual environment label in softmax probability, with dimensionality \( K \).
>
> Experiments
>
> 1. The ambiguity in the figure was due to an error during its creation. This has been corrected in the revised version.
>
> 2. The experiments were conducted as an average over multiple subject-independent folds, following dataset splits used in previous works and adhering to multi-fold cross-validation. However, we note that similar practices were not mandatory in all prior studies.
>
> 3. We clarify that experiments without a temporal domain tokenizer, such as InfoNCE and HSIC in Table 1, were included. However, the CIFT method inherently relies on this tokenizer, making it inseparable.
>
> 4. Numbers and classes were provided in the initial version based on prior works, with clear specifications and citations. Further refinements have been added in the revised version.
>
> 5. "SF" specifically refers to seizure forecasting. Table 1 reports OoD performances for both seizure forecasting and detection tasks.
>
> Wording, Grammar and Organization
>
> We have refined the wording, grammar, and overall organization in the revised version. Additionally:  SEEG refers to stereo-electroencephalography.  PEFT stands for "parameter-efficient fine-tuning."  ERM means "empirical risk minimization," which involves training without any OoD generalization algorithms.

---

> > ### Comment · Reviewer_hmwf · 2024-11-29
> > **Response to Rebuttal - Weaknesses**
> >
> > **Quality of the Experiments**
> >
> > > 1. The experiments are conducted following standard practices in EEG foundation model research, where we report the mean performance across multiple test sets. Some baseline pre-trained checkpoints were unavailable due to intellectual property and privacy concerns, which restricted our ability to conduct statistical tests.
> >
> > In my opinion, the field of EEG foundation model research is still in its infancy. The critical research question is whether foundation models can improve performance upon domain-specific models.
> > Within each domain (e.g., seizure prediction, motor imagery, sleep staging) the community has established appropriate evaluation criteria and protocols.
> > Since foundation models also require tuning of some parameters (e.g., classification heads) to a particular downstream task (e.g., application domain), established domain-specific evaluation procedures have to be be applied.
> >
> > In the specific-case of the EEG domain, the downstream datasets are relatively small. So, even a simple classification head can easily overfit the data. That is why, I consider it imperative to report variability of the results, carefully present the evaluation approach (especially on how the data splits were generated), and ideally apply significance testing (e.g., a difference in a few percent points might not be statistically significant for a small dataset). Thus, without any indication about variability and the evaluation approach, the value of the presented results is negligible.
> >
> > Since the authors decided to not address this critical weakness, I will have to maintain my current score.
> >
> > > 2. We clarify that full-parameter and linear probing results were presented in Table 1 as ERM-Full and ERM-LP, respectively. Moreover, choice of rank has significant influence on the performance. Given limited computational budget, larger rank leads to better performance. Ablation of different adapters in CIFT was presented in the appendix.
> >
> > Thanks for the clarification. Unfortunately, the presentation in the revised manuscript is still not clear. Please also adjust the corresponding paragraph in the revised manuscript.
> >
> > > 3. Regarding the visual discrepancy between the performance reported in Table 1 and Figure 5, it is not due to differences in the underlying data, but rather an artifact in the visual representation in Figure 5, which arose from the way the graph was rendered in Python using Matplotlib.
> >
> > Thank you for sharing your explanation. Since this issue has not been resolved in the revised manuscript, it remains open for me.
> >
> > > We have refined the details in the revised version.
> >
> > Thank you for partially resolving this weakness in the revised manuscript. At least you state that you used the physionet MI dataset for your motor imagery experiments. Still, it is not clear which evaluation scheme you used to generate the splits, and whether you used all classes or a subset.
> >
> > **Clarity of CIFT**
> >
> > > We have addressed and clarified the ambiguities in the CIFT section in the revised version, providing specific clarifications in response to the reviewer’s points as follows.
> >
> > > 1. We clarify that the [CLS] token is not used in CIFT or any other OoD generalization algorithms presented in this work. This has been corrected in the revised version.
> >
> > Thank you for the clarification. Still, this inconsistency between the text and Figure 1 remains in the revised manuscript.
> >
> > > 2. (\rho) is defined as an auxiliary function for estimating virtual environment labels for each sample. It is not used during the inference stage of CIFT. The additional parameters (\eta), introduced by (\rho), are implemented via a two-layer MLP, as already noted in the main text.
> >
> > Thanks for providing this additional information. Please also add it to the revised manuscript!
> >
> > > 3. We highlight that the dimensions for the contrastive fine-tuning loss were provided in the corresponding subsection.
> >
> > After fixing typos and the equations, it is indeed easier to follow the definition of CIFT. Still, the dimensions of $\hat{U}$, $\hat{V}$ and $\hat{e}$ are not defined.
> >
> > > 4. We clarify that (\rho) is solely used to estimate virtual environments (\hat{e}), not class labels (\hat{y}). Furthermore, the number of virtual environments (K) is unrelated to the number of classes, and this distinction is now clearly articulated.
> >
> > The error in equation (8) introduced the misconception. I noticed that you fixed it in the revised manuscript.

---

> > > ### Comment · Reviewer_hmwf · 2024-11-29
> > > **Response to Rebuttal - Questions**
> > >
> > > **Preliminaries**
> > >
> > > > It is defined as a sample and its label belonging to environment ( e ) within a potential set of environments.
> > >
> > > Thank you for the clarification. I note that the issue is not yet resolved in the revised manuscript.
> > >
> > > > Relevant works were cited in the OoD baselines section, as well as additional related work section in the appendix in the initial version. Additional citations have been included in the revised version for completeness.
> > >
> > > Thank you for providing references for further reading.
> > >
> > > **Methods**
> > >
> > > > We clarify that for a 4-second EEG sample at a sampling rate of 256 Hz, raw wavelet features are downsampled to ( 64 \times 256 ), with a patch size of ( 8 \times 8 ) adopted.
> > >
> > > Thank you for the clarification. I note that the issue is not yet resolved in the revised manuscript.
> > >
> > > ---
> > >
> > > I note that my question 3 in about the methods section - **which probabilistic model did you assume in equation (4). If you used an isotropic Gaussian model, the NLL loss reduces to the classical MSE loss. Please clarify**  - remains unaddressed.
> > >
> > > ---
> > >
> > > > The issue has been corrected in the revised version.
> > >
> > > > We highlight that (\hat{e}) represents the predicted virtual environment label in softmax probability, with dimensionality ( K ).
> > >
> > > Thank you for the clarification. I note that the issue is not yet resolved in the revised manuscript.
> > >
> > >
> > > **Experiments**
> > >
> > > > The ambiguity in the figure was due to an error during its creation. This has been corrected in the revised version.
> > >
> > > As pointed out above, this issue has not been resolved in the revised manuscript that is available on openreview.
> > >
> > > > The experiments were conducted as an average over multiple subject-independent folds, following dataset splits used in previous works and adhering to multi-fold cross-validation. However, we note that similar practices were not mandatory in all prior studies.
> > >
> > > Further details regarding data split generation are mandatory (i.e., were the data split randomly at the level of trials? Were the folds generated to test generalization across days/subjects?, ...). As pointed out above, reporting merely the average is not enough.
> > >
> > > > We clarify that experiments without a temporal domain tokenizer, such as InfoNCE and HSIC in Table 1, were included. However, the CIFT method inherently relies on this tokenizer, making it inseparable.
> > >
> > > At this stage it is too late to ask for additional experiments. Still, it would be interesting to investigate any structure in the learned masked $m_V$ and $m_U$. Interesting questions for future work could be: does the model rely more on features in U or in V? How diverse/redundant is the information in U and V?
> > >
> > > > Numbers and classes were provided in the initial version based on prior works, with clear specifications and citations. Further refinements have been added in the revised version.
> > >
> > > I could not find this information in the original manuscript nor in the revised version.
> > >
> > > > "SF" specifically refers to seizure forecasting. Table 1 reports OoD performances for both seizure forecasting and detection tasks.
> > >
> > > The confusion still remains. What does `SF` in the `Category` column in Table refer to? Based on your response, the most intuitive wording to categorize the methods in the first 4 rows is `ERM`.

---

### Official Review · Reviewer_XaU7 · 2024-11-01

**Soundness:** 3
**Presentation:** 3
**Contribution:** 3
**Rating:** 8
**Confidence:** 4

**Summary:**

The paper presented a new EEG foundation model, which used the wavelet-based spectral tokenization. The model performance were systematically evaluated on many EEG task. And the new foundation model focused on the Out-of-Distribution generalizability in EEG analysis. The work was technically sound and solid. Overall, I like the paper and the work would be meaningful to the field.

**Strengths:**

The proposed EEG foundation model not only used the temporal tokenization that were used in previous studies, but included new wavelet-based spectral tokenization. Spectral features in EEG were usually very useful and stable for EEG analysis. Introducing the new module was a good idea.

**Weaknesses:**

More experimental results on diverse EEG tasks should be well organized in the main text. The main text only included the results of seizure detection and forecasting.

**Questions:**

1.  What were the importance of the two tokenization: the temporal tokenization and wavelet-based spectral tokenization. Did the authors have the ablation experiments on two modules? How much improvement did the spectral tokenization have for the model?
2. How was the out-of-distribution generalizability defined in the paper, different subjects, different tasks, or different data centers? I think the study would be strengthened to include more systematic experiments and analysis for the out-of-distribution generalizability.
3. Why the EEG spectrogram in Figure 2 had up to 128 Hz frequency? We usually analyze up to 50 Hz spectrogram for scalp EEG. What were the frequency resolution and time resolution of spectrograms as the inputs to the spectral tokenization?

---

> ### Author Response · Authors · 2024-11-26
>
> We thank the reviewer for the helpful feedback.
>
> Weaknesses
>
> (1)	We clarify that we also discussed performance on sleep staging, motor imagery and brain decoding tasks in the main text, with detailed results in the appendix.
>
> Questions
>
> (1)	The Temporal Tokenizer is implemented using a 1D CNN, trained during the fine-tuning stage as part of the CIFT method, which is the primary OOD approach proposed in this work. Spectral tokenization, on the other hand, is utilized throughout both pretraining and fine-tuning stages, employing a frozen spectral tokenizer implemented via a VQGAN. We included ablation experiments in the appendix, which demonstrate significant performance improvements when both tokenizers are used together, as is done in CIFT.
>
> (2)	In this work, our primary focus is not on addressing discrepancies between different OOD tasks. Instead, we aim to propose a unified solution that demonstrates improved performance across distinct OOD tasks, emphasizing generalizability and robustness.
>
> (3)	High-frequency activities, such as gamma-band and high-frequency oscillations (HFOs) above 50 Hz, are indeed crucial for understanding cognitive functions and seizure development. Our approach does not impose specific frequency or time resolution constraints on the input data. For consistency, we adopted a 4-second segment length with a sampling rate of 256 Hz as the default configuration throughout our experiments.

---

### Official Review · Reviewer_6GMv · 2024-11-04

**Soundness:** 2
**Presentation:** 2
**Contribution:** 2
**Rating:** 3
**Confidence:** 3

**Summary:**

To address the challenges faced by EEG models in out-of-distribution scenarios, this work proposes a foundation model that employs a semi-causal generative modeling approach during the pretraining phase. In the fine-tuning stage, the model utilizes the Contrastive Invariant Fine-Tuning method, integrating both time-domain and frequency-domain information for representation learning.

**Strengths:**

This work conducted pretraining on a large-scale EEG dataset, focusing on key issues within EEG modeling. The model incorporates both time-domain and frequency-domain representations of EEG signals.

**Weaknesses:**

1. The writing of this paper is somewhat disorganized, with the related work section lacking necessary summaries and connections. The introduction of the methodology is also lacking in coherence and emphasis, resulting in low readability.
2. While this work draws on advanced methods from various fields, there is insufficient discussion regarding the motivation and rationale for transferring these methods to the EEG domain, leading to a perception that the model is somewhat stitched together and lacks innovation.
3. In terms of experimentation, the validation primarily relies on a self-collected dataset, which diminishes its persuasive power. Furthermore, the main text contains too few experimental results, lacking essential ablation studies, parameter experiments, and visualization results, rendering the findings inadequate.
4. The comparative methods employed in this work primarily consist of algorithms related to out-of-distribution (OOD) scenarios. However, as this study serves as a foundation model pretrained on large-scale data, it is recommended that comparisons be made with other foundation models.

**Questions:**

1. The contribution titled "Spectrotemporal Integration of EEG Representation" may be insufficient as a distinct contribution point, as this conclusion lacks novelty.
2. It appears that many related works have employed Analytic Wavelet Spectral features; however, there is a lack of citations to relevant literature.

---

> ### Author Response · Authors · 2024-11-26
>
> We thank the reviewer for the helpful feedback.
>
> Weaknesses
>
> (1) While we agree that presentation could be further refined, we emphasize that the main contributions are clearly highlighted in the Introduction section of the manuscript.
>
> (2) We provided a comprehensive review of prior works in the Related Work section, with additional details included in Appendix F for further context.
>
> (3) A significant portion of our experiments was conducted on publicly available datasets, including seizure detection, seizure forecasting, sleep staging, motor imagery, and brain decoding. Detailed results are provided in the Appendix. For the self-collected datasets, we have made them accessible via an anonymous link. Ablation studies are also thoroughly presented in the Appendix.
>
> (4) We clarify that we compared our results with several state-of-the-art foundation models in Tables 3, 4, 9, 10, and 11, demonstrating the competitiveness and robustness of our approach.
>
> Questions
>
> (1) We clarify that the spectrotemporal integration of EEG representation is not only distinct but also critical, as it underpins the effectiveness of our CIFT method. By leveraging both spectral and temporal tokenization, our approach enables robust domain-invariant representations essential for EEG's non-stationary nature. Widespread experiments have shown that CIFT achieved substantial performance gains by integrating these features in an environment-aware manner, highlighting its novelty and transformative impact across tasks.
>
> (2) The analytic wavelet transform is a well-established technique widely discussed in college-level signal processing textbooks, and relevant references have been included in the revised version.

---

### Meta-Review · Area_Chair_wmm2 · 2024-12-15

**Metareview:**

The EEG foundation model research is still in its infancy, so the topic itself is timely. This paper claims a few things in a single paper, including pre-training an EEG foundation model, corresponding parameter-efficient fine-tuning, and domain generalization. The strength in this paper is in pre-training a model on large scale EEG data, leveraging both time-domain and frequency-domain representations of EEG signals. One reviewer was very positive on this paper. However, the rest of reviewers have concerns in the presentation, lack of novelty, and experiments.
Therefore, the paper is not recommended for acceptance in its current form. I hope authors found the review comments informative and can improve their paper by addressing these carefully in future submissions.

**Additional Comments On Reviewer Discussion:**

During the discussion period, there was no change at all. Thus, all reviewers stood by their original decision.

---

### Decision · Program_Chairs · 2025-01-22

Reject